# *Capsicum chinense* Jacq. fruit plays an immunomodulatory role in cytokine attenuation and DNA damage protection

**Srabonti Saha, Md. Altaf Hossain, Humayra Ferdousi, Jobaier Ibne Deen,**
**Akhlak Chowdhury, Md. Mannan, Md. Asif Nadim Khan, Md. Atiar Rahman***

Department of Biochemistry and Molecular Biology, University of Chittagong, Chittagong, Bangladesh

* atiar@cu.ac.bd

## Abstract

*Capsicum chinense* (*C. chinense)* Jacq., recognized for its bioactive compounds, has attracted interest due to its possible immunomodulatory and DNA-damage-protective effects. This study aimed to assess the immunomodulatory, anti-inflammatory, and DNA protection abilities of organic extracts (methanol, ethyl acetate, and petroleum ether) from *C. chinense.* The immunomodulatory effects were evaluated in Long Evans rats induced with SRBC. At the same time, the anti-inflammatory potential was investigated in LPS-stimulated RAW264.7 macrophages by quantifying pro-inflammatory mediators such as COX-2, iNOS, TNF-α, IL-β, and IL-6. Protective activity against DNA damage was assessed using a method that induces damage with a Fenton reagent. Cytotoxicity was tested on the Hela cell line to gauge the cellular effects of the extracts. The results demonstrated that higher doses (200 mg) of *C. chinense* methanol extract inhibited immune responses, whereas lower doses helped restore them. The extracts significantly decreased pro-inflammatory mediators and cytokines in LPS-activated macrophages. Both petroleum ether and methanol extracts showed higher cytotoxicity against Hela cells compared to the ethyl acetate extract. The protein levels recorded in the serological parameters were 5.74 ± 0.19, 5.36 ± 0.11, 5.74 ± 0.26, 6.02 ± 0.16, 6.18 ± 0.18, and 6.22 ± 0.20 gm/dL for NC, DC, ML, MLMExCC100, MLMExCC200, and MExCC100, respectively. These findings suggest that *C. chinense* extracts possess strong immunomodulatory effects and potential protection against DNA damage, supporting their therapeutic application in regulating the innate immunity system.

## 1. Introduction

Immunomodulation refers to any alteration in the immune response, including the induction, expression, amplification, or inhibition of any component or stage of the immunological response. In some people, including immunocompromised patients, the immune response must improve. Whereas suppression of immune response is thought for others, such as transplant recipients, patients with autoimmune diseases, and allergic and inflammatory disease patients [1]. Apart from being specifically stimulatory or suppressive, specific agents normalize or modulate pathophysiological processes and are hence called 'immunomodulatory

**Data availability statement:** All data are placed within the manuscript.

**Funding:** The author(s) received no specific funding for this work.

**Competing interests:** The authors have declared that no competing interests exist.

**Abbreviations:** NC, Normal control group; DC, Disease control group; MExCC, Methanol extract of Capsicum chinense; EAExCC, Ethyl acetate extract of Capsicum chinense; PEExCC, Petroleum ether extract of Capsicum chinense; ML, Milk; MLMExCC, Milk with methanol extract of Capsicum chinense; SRBC, Sheep red blood cells; DTH, Delayed-type hypersensitivity; PBS, Phosphate-buffered saline; FBS, Fetal bovine Serum; $H_2O_2$, Hydrogen peroxide; COX 2, Cyclooxygenase 2; iNOS, Inducible nitric oxide synthase; IL-6, Interleukin-6; IL-1β, Interleukin-1β; TNF-α, Tumor necrosis factor-alpha; LPS, Lipopolysaccharide; NF-κB, Nuclear factor kappa B; MAPK, Mitogen-activated protein kinase

agents [2]. Increasing interest surrounds the modulation of our immune response through various bioactives to assist in alleviating specific diseases. Notably, medicinal plants present a viable alternative to conventional chemotherapy for many conditions. This approach is particularly beneficial for strengthening our body's defense mechanisms, especially in cases of compromised immune responses or when targeted immunosuppression is required, such as in autoimmune conditions. Many plants and their isolated constituents have been shown to potentiate immunity and exert anti-inflammatory, anti-stress and anti-cancer effects by modulating the immune functions. However, it's important to note that medicinal plants and their derived products haven't quite made a big splash yet in the development of adjuvants for vaccination programs or immunosuppressants. This is largely due to a lack of emphasis on plant-based immunomodulators, which hold great potential. Fortunately, some plant products are being investigated for immune response modifying activity and plethora of a plant-derived materials (proteins, lectins, polysaccharides, etc.) are manifested to stimulate the immune system [3]. Some of the plants including *Asparagus racemosus*, *Azadirachta indica*, *Ocimum santum, Panax ginseng, Polygala senega, Tinospora cordifolia, Viscum album*, are well established for their immunomodulatory activity [4,5]. However, strong surveillance of newer sources of plant-based immunomodulators is still inexorable for prospective therapeutic approaches. An extensive study for the development of new methods needs to be done to create innovative immunomodulators from plants.

The Naga Chili, or *C. chinense* Jacq., is the hottest chili pepper in the world, according to the Guinness Book of World Records (2006) [6], with a Scoville heat unit (SHU) rating ranging from 879,953 to 1,001,304. In Bangladesh, hillside orchards are renowned for their citrus fruits and Naga chillies. The Naga chili plant is a remarkable natural gift due to its pleasant aroma and taste. Originally known as Naga Morish or Dorset Naga, this crop was cultivated in the Brahmaputra floodplain of northeastern Bangladesh [7]. *Capsicum chinense* Jacq. has been used medicinally for centuries to treat a wide range of conditions. Several researchers have talked about how *C. chinense* Jacq. has been used in Chinese medicine for a long time and what the likely scientific basis is for such uses [8]. *C. chinense* Jacq. fruits have been used for a long time as a folk medicine to help restore muscle tone after hard physical work Hot infusion [9] of *C. chinense* Jacq. is used for aches and pains in the teeth and muscles. The use of *C. chinense* as an essential spice or spicy vegetable is common in the Indian subcontinent. Bioactive capsaicinoids from *C. chinense and* other hot chilis distinctively serve as anti-tumor, anti-cancer, antioxidant, and anti-obesity molecules [10]. The pharmacological importance of *C. chinense* necessitates the investigation of immunomodulatory and relevant biological functions of *C. chinense* and its products Numerous studies, both experimental and epidemiological, have investigated the potential of natural compounds to modulate the immune system and reduce inflammation [11]. The natural compounds possess the capacity to alter the expression of various pro-inflammatory genes, including multiple cytokines, lipoxygenase, nitric oxide synthases, and cyclooxygenase. Additionally, they exhibit antioxidant properties by protecting DNA damage which aids in the regulation of inflammatory signaling [12,13] This study aimed to evaluate the immunomodulatory functions of *C. chinense* organic extracts along with their anti-inflammatory and DNA damage protecting capacity.

## 2.  Materials and methods

### 2.1.  Chemicals and reagents

The necessary chemicals such as methanol, n-hexane, ethyl acetate, petroleum ether, tris, boric acid, agarose, ferric chloride, hydrogen peroxide, plasmid, chloroform, ascorbic acid, $Na_2EDTA$, hematoxylin (CAS No. 517-28-2), xylene, and eosin were procured from

Sigma-Aldrich (St Luis, MO). All the chemicals used in the experiment were of analytical grade.

## 2.2. Collection and identification of *Capsicum chinense* Jacq

*Capsicum chinense* Jacq. (*C. chinense* Jacq.) green fruits (10 kg) were obtained from a local farmer in the Reazuddin Bazar area of Chattogram (GPS 22.3365°N, 91.8303°E). The fruit was identified and botanically assigned by Dr. Shaikh Bokhtear Uddin (bokhtear@cu.ac.bd) who is a plant taxonomist and Professor of the department of Botany, University of Chittagong. Briefly, the fruits description was made including the plant habit (tree, shrub, vine, herb), height, growth form, color, bark, branching, leaf orientation, or general volume/size/spread of a plant. Detailed location of the *C. chinense* plant, origin of material, habitat, frequency (whether the plant is rare or common), herbarium sheet preparation (pressing and drying the sample, labelling, mounting), comparing with the published plant descriptions, illustrations and photographs to identify the herbarium specimens. The Departmental herbarium received the voucher specimens, which are identified by accession number (ss020619-531).

The quality of fruits was ensured to be free from adulteration. The major criteria were the colour, appearance, texture, flavour, weight, and volume. No spotted/rotten/unhealthy/infectious fruits were accepted. Very fresh and natural fruits were accepted for our research.

## 2.3. Preparation of extracts

The collected *C. chinense* Jacq. samples were shade-dried and mechanically crushed into a coarse powder (Sonipat, Heavy Duty10 Electric Dry Masala & Herbs Grinder, Swing Type 2300W Haryana, India). The resulting powder was defatted using n-hexane and was extracted using petroleum ether, ethyl acetate and methanol for 3 days respectively.

The filtrates were then dried at 40°C in a rotatory evaporator (RE200, BIBBY Sterilin Ltd. Staffordshire, UK) at medium pressure (100 mbar), and the resulting crude extracts were stored in the refrigerator at 4°C until further use. The sample was henceforth designated as methanol extract of *C. chinense* (MExCC); ethyl acetate extract of *C. chinense* (EAExCC), and petroleum ether extract of *C. chinense* (PEExCC) [14].

## 2.4. *In vivo* Immunomodulating activity of *C. chinense* Jacq

**2.4.1. Experimental design.** Long Evans rats of both sexes weighing 90–120 g were utilized for immunological investigations acquired from the International Centre for Diarrheal Disease Research, Bangladesh (icddr,b, Dhaka). The animals were then randomly categorized into 7 groups, with 6 animals in each group, depending on the concentration of the drug given to them. The animals were housed in cages with rice husks serving as their bedding. The animals were housed in transparent polycarbonate rat cages measuring 430 mm in length, 290 mm in width, and 201 mm in height. The environment maintained a temperature of (24 ± 2) °C and a humidity of (55% ± 5), along with a 12-hour light-dark cycle. They had unlimited access to their standard pellet diet water. The rats were acclimatized for a week. Animal handling and cares were confirmed under an ethical guideline endorsed by the Institutional Ethical Committee of the Faculty of Biological Sciences of the University of Chittagong (Ethical approval no. AERB-FBSCU-20230605-(1)).

The animals were grouped and treated with MExCC as follows:

Group I (NC) - Normal healthy control rats treated with water.

Group II (DC): Disease control rats treated with antigen.

Group III (ML) – Rats treated with milk.

Group IV (MLMExCC 100) - Rats treated with milk + 100 mg/kg bw of MExCC.

Group V (MLMExCC 200) - Rats treated with milk + 200 mg/kg bw of MExCC.

Group VI (MExCC 100) - Rats treated with 100 mg/kg bw of MExCC

Group VII (MExCC200) - Rats treated with 200 mg/kg bw of MExCC

The dose volume was calculated to be not more than 1 mL of drug preparation per animal. Control animals received 1 mL of water [15].

**2.4.2. Immunization.** Blood samples were collected from juvenile, adult, and robust sheep at Chittagong Veterinary and Animal Science University (CVASU). A sterile Alsever solution was prepared by combining blood with 2g of dextrose, 0.8 g of trisodium citrate dehydrate, 0.055 g of citric acid monohydrate, and 2.1 g of sodium chloride in a 1:1 ratio. The red blood cell underwent centrifugation at 1,600 g for 5 minutes before settling at the bottom of the test tube. The upper layer (plasma) and the intermediate layer were subsequently discarded. The red blood cell pellet was retained at the bottom of the centrifuge tube. The red blood cell pellet was subsequently resuspended in sterile phosphate-buffered saline (PBS) at pH 7.2. Subsequently, the centrifugation and washing procedures were repeated 3 times to eliminate residual plasma proteins and other contaminants. Prepare a diluted suspension of sheep red blood cells (SRBC) at a concentration of 1% in phosphate-buffered saline (PBS) for immunization. The SRBC quantity was calibrated to $0.5 \times 10^9$ cells/ml. All animals received the initial dose (1 mL) of immunization on the 7th day, utilizing a 1 ml syringe containing $0.5 \times 10^9$ cells/ml of SRBC. The booster dosage was administered on the 11th day, utilizing an equivalent quantity of fresh red blood cells as in the prior booster dose [16].

**2.4.3. Body weight and lymphoid organ weight.** After 14 days of therapy, the animals were anesthetized with an inhalation anesthesia using 5% of isoflurane (USP, 1349003) for approximately 5–7 min and decapitated when fully sedated, as measured by a lack of active paw reflex. Before sacrificing, their body weights were measured, and their hearts were punctured with a needle using a 21-gauge needle so that blood could be drawn. Several diagnostic procedures, including total hemocyte count and differential count, were carried out using the collected blood sample. In addition, the serum was produced to analyze the serum's albumin and globulin levels. Following the collection of blood, the animals were sacrificed, and the weight of organs such as the liver, thymus, and spleen was noted. Lymphoid tissues, such as the spleen, the thymus, the liver, and the kidney, were preserved in phosphate-buffered formalin so that histological examinations could be carried out later as a continuation of this line of study

**2.4.4. Determination of total serum protein and albumin: Globulin ratio.** This was calculated using the Biuret technique, and the absorbance was measured using an ultraviolet spectrophotometer at 540 nm [17].

**2.4.5. Total leukocyte count.** Total leukocyte was counted by WBC diluting fluid using a Hemocytometer [18].

**2.4.6. Differential count of white blood cells [19].** The blood smear was made on a fresh glass slide. After 3 min in methanol, the smear was dried and ready to be examined. The slide was air-dried, and then submerged in Field's solution B for 5 s. After that, it was washed and hung up to dry. After 15 s, Field's solution A was used to stain the sample. The stained slides were then washed and dried. The stained slides were examined under the microscope and the resolution was 400X.

**2.4.7. Hypersensitivity reaction.** On the day of the final treatment day, all the animals were sensitized by injecting 0.15 ml of $0.025 \times 10^9$ SRBC/ml subcutaneously into

the subplanter right hind paw using an insulin syringe. This was done on the day that the treatment was finished. Then the thickness of the rats' footpad was measured using a spheromicrometer (0.01 mm pitch) after 4hs, 8hs, and 24hs [16].

**2.4.8. Histopathological analyses.** Rat spleen and thymus specimens were removed, washed with sterile saline to remove any blood stains, stored in 10% buffered formalin, and then embedded in paraffin wax. The inserted tissue blocks were divided into 5 μm layers using a microtome device. Hematoxylin and eosin (H&E) were used to stain dewaxed sections. Using a light microscope, histopathological analyses were performed [14].

## 2.5. Anti-Inflammatory activity against LPS-stimulated RAW264.7 macrophages

**2.5.1. Cell lines and culture condition.** RAW264.7 macrophage cells were acquired from the ATCC (American Type Culture Collection) and grown in Dulbecco's modified Eagle's medium (DMEM), which included 10% (v/v) fetal bovine serum (FBS) and 1% (v/v) P/S at 37°C in a humidified environment containing 5% carbon dioxide.

**2.5.2. Morphological study using an inverted light microscope.** The morphological investigation was carried out according to [20]. Briefly, $10^5$ cells were plated into each well of a six-well plate, and then the plate was placed in an incubator for 24 h to allow the cells to connect. After that, the medium was withdrawn, and a new medium containing treated samples at their $IC_{50}$ concentration was added. After incubation of 72 h, morphological changes were studied under a light microscope. (Olympus, Tokyo, Japan).

**2.5.3. RT-qPCR for analysis of gene expression.** Using TRIzol reagent (Invitrogen), total RNA was extracted from both controls (DMSO alone) and treated (with 1 μg of LPS) RAW 264.7 macrophages, and the amount of RNA extracted was determined using a NanoDrop spectrophotometer. Purified total RNA (around 1 μg) was reverse-transcribed into cDNA in a final volume of 10 μL using a High-capacity RNA to cDNA Master Mix kit (Applied Biosystem), as per the manufacturer's instructions. Using the freshly synthesized cDNA as a template, RT-qPCR was conducted using SYBR qPCR mix (Nippon Gene) according to a standard technique [21]. Primer sequences (Table 1) used to quantify genes involved in inflammation are listed. The ΔCt values were calculated and the standard deviation of the Ct values of the ΔCt values was also calculated, which were used to determine the ΔΔCt values. And then, all data was collected and calculated as $2^{-\Delta\Delta Ct}$.

**Table 1. Forward and Reverse Primer of involved in inflammation.**

| TNF-α | Forward: ACG GCA TGG ATC TCA AAG AC<br>Reverse: GGT CAC TGT CCC AGC TT |
|---|---|
| iNOS | Forward: GTG GTG ACA AGC ACA TTT GG<br>Reverse: GGC TGG ACT TTT CAC TCT GC |
| IL-1β | Forward: GAG TGT GGA TCC CAA GCA AT<br>Reverse: CTC AGT GCA GGC TAT GAC CA |
| IL-6 | Forward: AGT TGC CTT CTT GGG ACT GA<br>Reverse: GCC ACT CCT TCT GTG ACT CC |
| COX-2 | Forward: TCC TCC TGG AAC ATG GAC TC<br>Reverse: TGA TGG TGG CTG TTT TGG TA |
| GAPDH | Forward: TGC TCG AGA TGT CAT GAA GG<br>Reverse: TTG CGC TCA TCG TAG GCT T |

## 2.6. DNA damage protecting activity

**2.6.1. Preparation of reagents.** The reagents were produced in accordance with the description provided by Thangara [22]. The following procedures need to be carried out to produce a solution of 10X TBE stock that is 1L in volume: To complete the operation, 108 gof Tris and 55 g of boric acid are dissolved in 800 mL of distilled water. Add 40 mL of a solution that has a pH value of 8.0 and contains 0.5 M of $Na_2EDTA$. Make sure that the volume is set to a value of 1L. It is necessary to thoroughly combine 50 mL of a 10-fold concentrated Tris-Borate-EDTA (TBE) stock solution with 950 mL of sterile water to make a solution of 0.5X TBE running buffer with a volume of 1L.

**2.6.2. Agarose gel.** First, weigh and transfer 0.4 g of agarose (0.8 percent gel) into a conical flask. Agarose was subsequently heated to achieve dissolution in 50 mL of 1x TBE running buffer. Occasionally stir the mixture to ensure uniform distribution. Ethidium bromide was incorporated into the molten gel at a concentration of 0.5 g/mL after the temperature had decreased to approximately 60°C. Then inserted a suitably sized and shaped comb into the mold. After allowing the gel to cool, pour it into the gel mold. The gel solidified and can be utilized for sample loading once the comb is removed and the gel box is prepared.

## 2.7. DNA damage protecting assay

The protective effects exhibited by different concentrations of plant extract in protecting pBR322 plasmid DNA against the detrimental impact of hydroxyl radicals generated by Fenton's reagent were assessed using the DNA nicking test as previously published by Golla and Bhimathati in 2014 with some alterations [23]. The experimental solution consisted of 3µL of plasmid DNA, and 5 µL of Fenton's reagent (30 mM $H_2O_2$, 50 µM Ascorbic acid, and 80 µM $FeCl_3$). Subsequently, various concentrations of extract (0, 20, 40, 60, 80, 100 µg/mL) were added to the mixture. The final volume of the mixture was adjusted to 20 µL by adding double-distilled water to each PCR tube. One test tube contained plasmid and Fenton's reagent, but no extracts were kept as a negative control. When compared to the negative control of extract plus Fenton's reagent, positive control was plasmid alone. After that, the reaction mixture is allowed to be at 37°C for 30 min. Each test tube was then diluted with bromophenol blue dye (0.25% in 50% glycerol) after incubation. Thereafter, a precast agarose gel (0.8%) was loaded with all the tubes and placed in electrophoresis equipment for DNA fragment separation at 90 V for 1 h using ethidium bromide labelling. Running buffer was a 1x TBE solution that was put into the tank and used to submerge the gel in the buffer. DNA bands were finally seen in the gel. Sample and plasmid alone resulted in distinct bands. UV microscopy of DNA stained with ethidium bromide (EtBr) revealed bands of DNA indicative of fragments formed from radical damage after being exposed to hydrogen peroxide ($H_2O_2$). When samples were introduced to the plasmid and Fenton's reagent, the antioxidant reduced the damage of hydroxyl radicals in a dose-dependent manner. The BIO-VISION V17.06 Gel Documentation system was used to examine the bands.

## 2.8. Cytotoxic effect of *C. chinense* Jacq.

Briefly, DMEM (Dulbecco's Modified Eagles' medium) containing 1% penicillin-streptomycin (1:1) and 0.2% gentamycin and 10% fetal bovine serum (FBS) was used to maintain the cell lines BHK-21, a baby hamster kidney fibroblast cell line, HeLa, a human cervical carcinoma cell line, and Vero, an African green monkey kidney epithelial cell line. BHK-21 Cells ($3 \times 10^4/200$ µL), HeLa Cells ($4.0 \times 10^4/200$ µL), and Vero cells ($3 \times 10^4/200$ µL) were seeded onto a 48-well plate and incubated at 37°C + 5% $CO_2$. The next day, 50 µL of the sample

(filtered) was added to each well. Cytotoxicity was examined under an inverted light microscope after 48h of incubation. Duplicate wells were used for each sample.

## 2.9. Statistical analysis

All information is shown as mean ± SD of 6 animals. The data were assessed using the t-test, one-way ANOVA, two-way ANOVA, and Tukey's multiple range post hoc tests in the statistical program Graph-Pad Prism v8.4.3. The results were deemed to be substantially different at the levels of *P ≤ 0.05, **P ≤ 0.01, and ***P ≤ 0.001.

# 3. Results

## 3.1. Immunomodulatory activities

### 3.1.1. Effect of *C. chinense* Jacq. extracts on body and organ weight.
The effects of *C. chinense* on the body and organ weights are summarized in Tables 2 and 3. All groups, except the highest extract dosage group, gained considerably more weight compared to the NC group. The final body weight of the milk-treated group was somewhat higher compared to the NC group. However, the body weight of both MLMExCC100 and MLMExCC200 groups were quite similar. The MExCC100 group had a higher final body weight than the other groups. The highest dose was found to have a minimal effect on the body weight than the other doses.

There was no statistically significant difference in the total weight of the livers among the groups. The same was true for the spleen, which showed no significant differences between

**Table 2. Effects of extracts on body weight.**

| Groups | Initial weight (G) | Final weight (G) | Weight change (G) |
|---|---|---|---|
| NC | 100 ± 6.72 | 152.66 ± 7.91 | 52.66 ± 0.84 |
| DC | 101 ± 5.44 | 133.5 ± 2.66** | 32.5 ± 1.96 |
| ML | 103.16 ± 8.01 | 166.16 ± 7.13 | 63.00 ± 0.62 |
| MLMExCC100 | 101 ± 8.53 | 144.66 ± 10.69 | 43.66 ± 1.53 |
| MLMExCC200 | 104.66 ± 5.39 | 144.16 ± 8.01 | 39.5 ± 1.85 |
| MExCC100 | 109.33 ± 6.74 | 163.16 ± 8.88 | 53.83 ± 1.51 |
| MExCC200 | 107.5 ± 8.09 | 133.33 ± 11.03** | 25.83 ± 2.08 |

Values are expressed as mean ± SD, where *n* = 6 animals were compared with one-way ANOVA, followed by Tukey's multiple comparison test.

**$p < 0.01$.

**Table 3. Effects of extracts on organs weight.**

| Groups | Liver weight(G) | Spleen weight(G) | Thymus weight(G) |
|---|---|---|---|
| NC | 5.98 ± 0.42 | 0.47 ± 0.02 | 0.47 ± 0.04 |
| DC | 6.60 ± 0.35 | 0.56 ± 0.06 | 0.4 ± 0.03 |
| ML | 6.14 ± 0.18 | 0.51 ± 0.04 | 0.5 ± 0.40 |
| MLMExCC100 | 6.00 ± 0.23 | 0.04 ± 0.45 | 0.39 ± 0.03 |
| MLMExCC200 | 5.66 ± 0.22 | 0.46 ± 0.05 | 0.37 ± 0.02 |
| MExCC100 | 6.04 ± 0.44 | 0.47 ± 0.03 | 0.44 ± 0.08 |
| MExCC200 | 6.61 ± 0.64 | 0.51 ± 0.06 | 0.23 ± 0.043*** |

Values are expressed as mean ± SD, where *n* = 6 animals were compared with one-way ANOVA, followed by Tukey's multiple comparison test.

***$p < 0.001$.

the treatment groups and the normal control group. The thymus weight achieved by the highest dose group was found to be significantly (P < 0.05) different from that of other groups while other doses had no significant differences (Table 3).

**3.1.2. Delayed type hypersensitivity test.** Fig 1 presented the effects of *C. chinense* extracts on the delayed type of hypersensitivity that was noticed for every group of SRBC-induced animals. A remarkable increase in volume was recorded for both doses. After 4 h of SRBC injection, the thickness of paw was recorded 19.8 ± 3.89%, 20.2 ± 4.76%, 34.0 ± 6.52%, 34.0 ± 5.48%, 34.0 ± 4.18%, 58.8 ± 18.59%, and 35.6 ± 5.85% respectively for NC, DC, ML, MLMExCC100, MLMExCC200, MExCC100, and MExCC200. After 24 h the thickness of paw was recorded 0.8 ± 1.09%, 18.8 ± 3.56%, 0.8 ± 1.09%, 2.40 ± 1.81%,.2.40 ± 2.88%, 5.40 ± 2.50%, 32.4 ± 4.50% for NC, DC, ML, MLMExCC100, MLMExCC200, MExCC100, and MExCC200. However, the effects of ML, MLMExCC100, MLMExCC200, MExCC100, and MExCC200 were not significant compared to the NC group (Fig 1).

**3.1.3. Hematological assays.** The total leukocyte counts, and differential leukocyte counts are presented in Fig 2A and 2B respectively. When compared to the NC group, total leukocyte count was observed to be elevated in the MExCC100 as well as in the ML, MLMExCC100, and

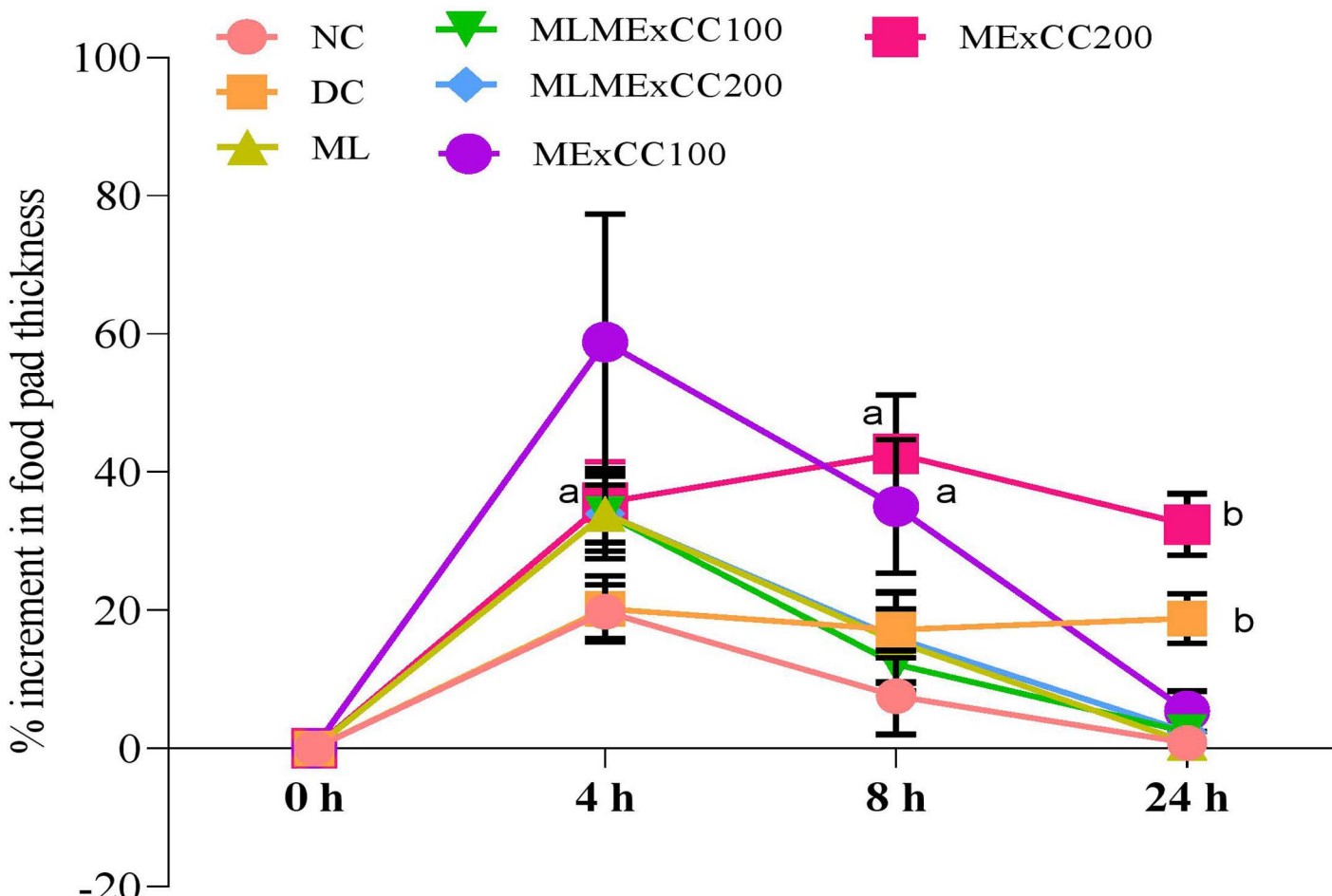

**Fig 1. The effects of *C. chinense* extracts on the delayed type of hypersensitivity of experimental animals (n = 6).** Animals fed with ML, MLMExCC and MExCC develop hypersensitivity. Values are expressed as mean ± SD for six animals in each group compared with Two-way ANOVA followed by Tukey's multiple comparison test.

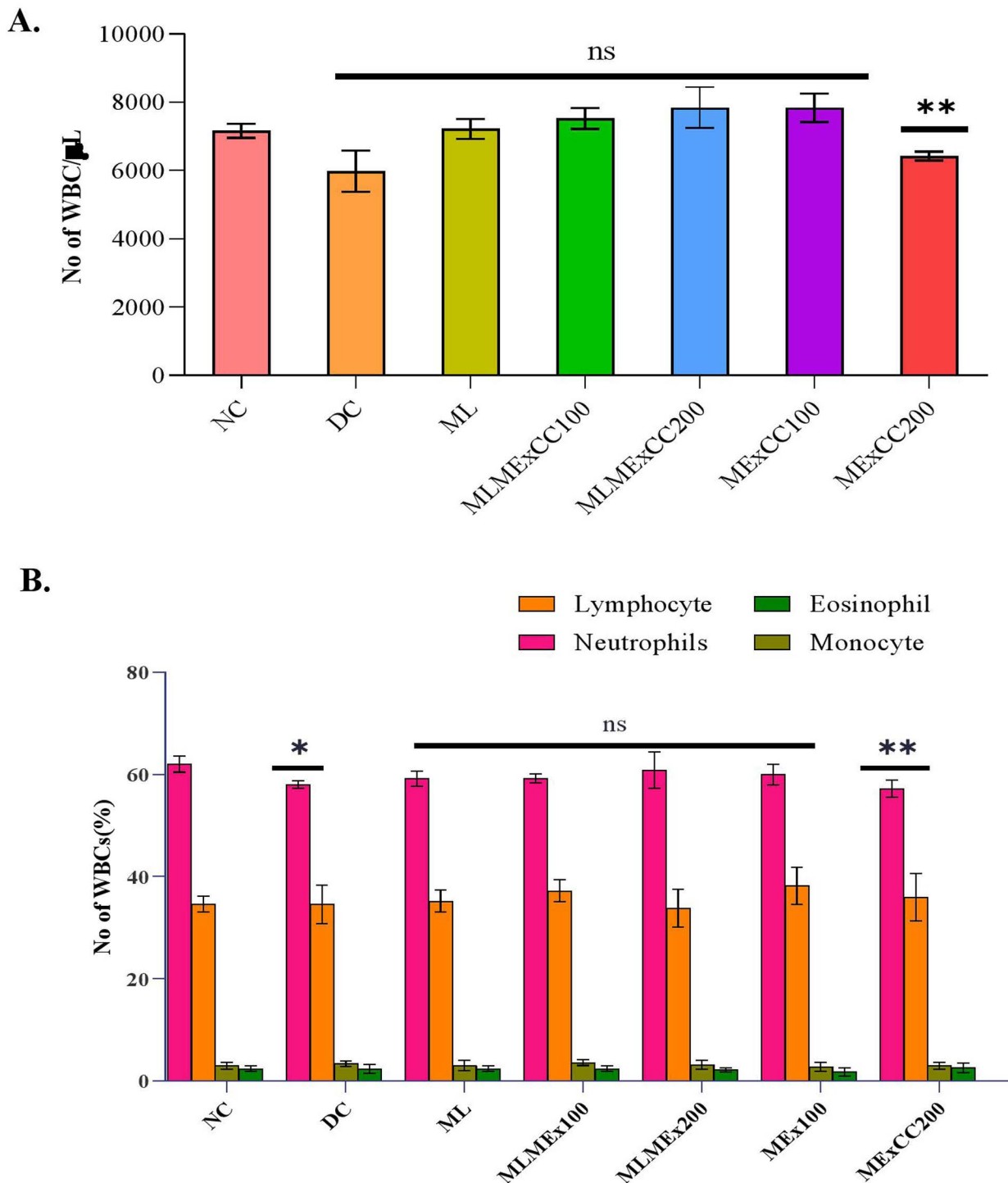

**Fig 2. (A). Total count; (B) Differential counts of WBCs are assessed.** Comparisons were made with normal control. Inference (Two-way ANOVA with Tukey's multiple comparison) to interpret significant difference. Data expressed as mean ± S.D where n = 6. *= p < 0.05 **= p < 0.01. ns=nonsignificant.

MLMExCC200. However, an increase in the extract concentration has resulted in a further reduction in the overall number of leukocytes. The percentage of neutrophils was found to be maximum in the MLMExCC200 group and in other extracts. The monocyte and eosinophil count for treatment groups were not changed significantly compared to the NC group.

**3.1.4. Serological parameter.** The serological parameters displayed in Fig 3 showed that the blood protein level for the extract at MExCC100 was found to be significantly higher than that of the other extracts. The protein level was recorded at 5.74 ± 0.19, 5.36 ± 0.11, 5.74 ± 0.26, 6.02 ± 0.16, 6.18 ± 0.18, and 6.22 ± 0.20 gm/dL for NC, DC, ML, MLMExCC100, MLMExCC200, and MExCC100 respectively. On the other hand, in the case of MExCC200, the quantity of serum protein (5.46 ± 0.19 gm/dL) decreased. Fig 3 also summarizes the albumin/globulin ratio for different treatments. The ratio was increased in all extracts; however, the highest dose was noticed to slightly reduce the ratio. Above all, the AG ratio for the extracts was insignificant compared to NC.

**3.1.5. Histopathological assay of spleen and thymus.** Histopathological examination of the normal control group's spleen revealed no abnormalities. The white pulp of the spleens of rats in the control group had clearly defined T cell regions that surrounded the central arteries and formed periarteriolar lymphoid sheaths (PALS). In addition to this, the follicles contain germinal centers that are actively undergoing mitosis. At the follicular edge, there was a mantle zone, and beyond that, there was an enormous marginal zone. In Fig 4A, the marginal sinus was also easily seen, and the white and red pulp between the border was shown in a manner that was obvious and clear. In Fig 4B, the effects of SRBC can be seen as a

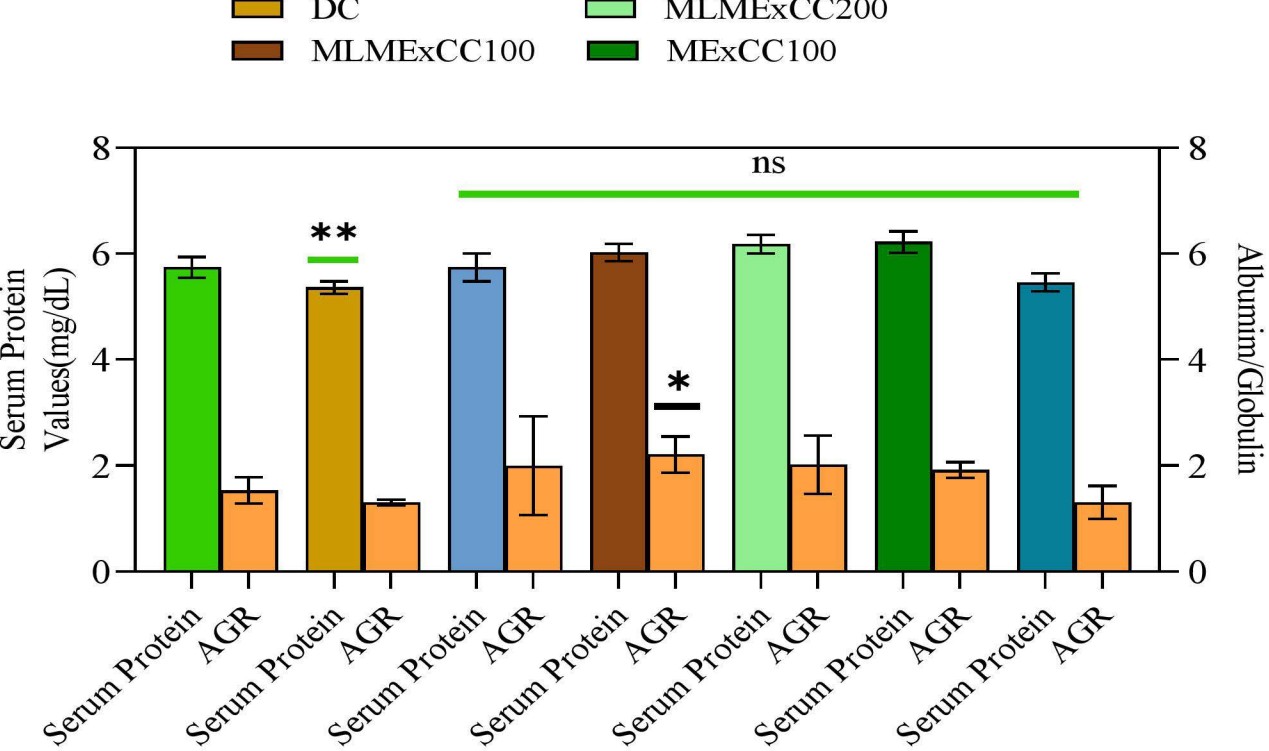

**Fig 3. Serum albumin/ globulin ratio is assessed.** Comparisons were made with normal control. Inference (Two-way ANOVA with Tukey's multiple comparison) to interpret significant differences. Data expressed as mean ± S.D where n = 6. **= p < 0.01, ns = nonsignificant.

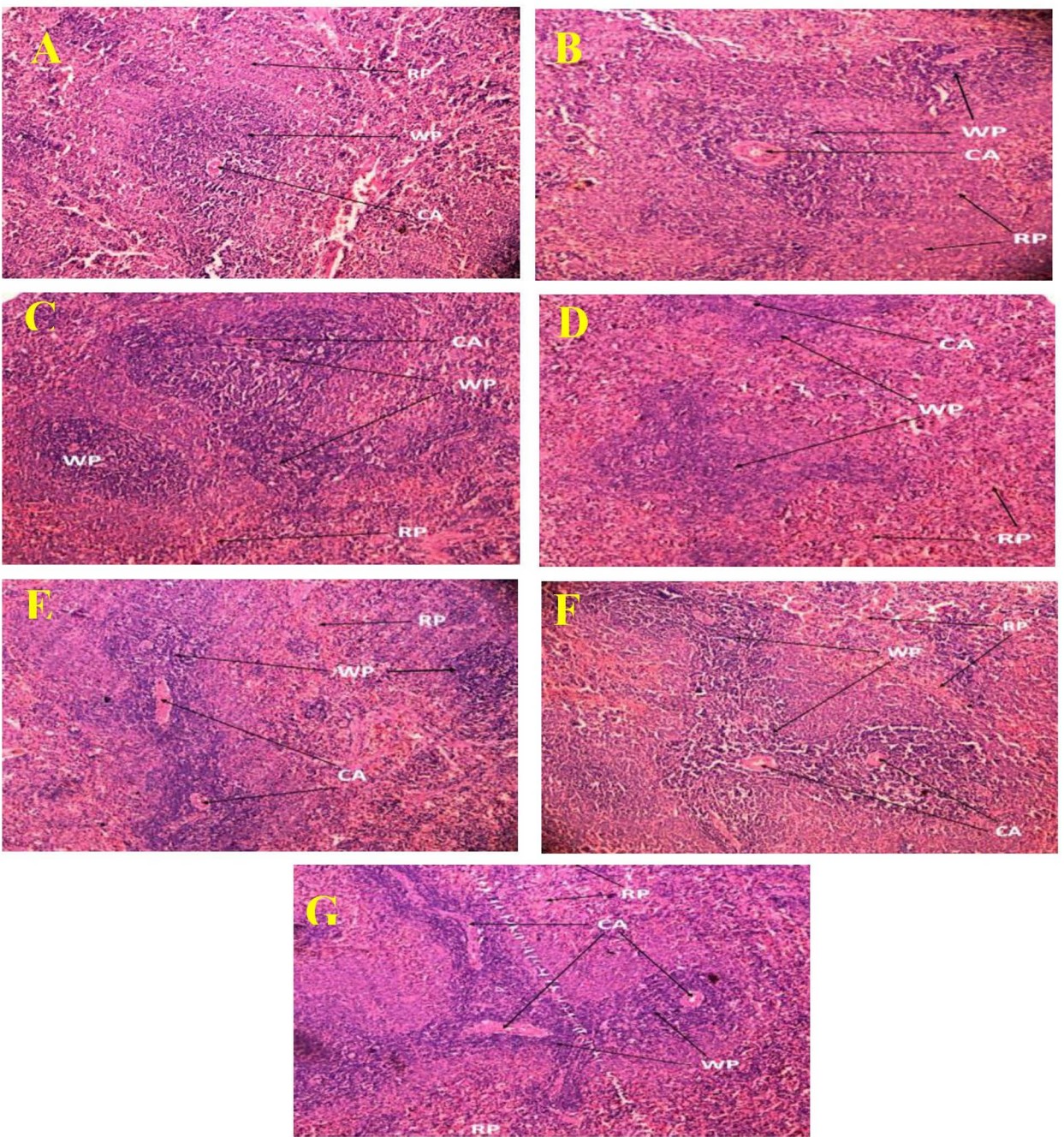

**Fig 4. Image of the spleen tissues from the experimental animal groups as seen through histopathology.** The white pulp and red pulp spleen cell are seen in the image (microscopic resolution: 10 × 40). Hematoxylin and eosin-stained rat spleen micrographs. The images displayed are the glomerulus slices counterstained with hematoxylin and stained with PAS.

dramatic reduction in the volume density of the lymph follicles and a large rise in the amount of red pulp. Additionally, the medulla includes more developed lymphocytes, conspicuous epithelial cells, admixed macrophages, Hassall's corpuscles, and dendritic cells. Although the disease control group's spleen exhibited an absence of full germinal centers in the white pulp of the spleen tissue, these centers were represented infrequently, had smaller diameters,

and were accompanied by a reduction in lymphocyte count. Many follicles contain germinal centers, and the boundary between the white pulp and the red pulp was well delineated in the ML-treated group's representation in Fig 4C. The spleen of the groups of rats to whom the MLMExCC100, MLMExCC200, and MExCC100 were given showed moderate white pulp with an increase in the number of reticuloendothelial cells in the red pulp (Fig 4D–4F). The administration of MExCC200 led to a discernible decrease of the volume density of the lymph follicles as seen in Fig 4G as well as a discernibly greater expansion of the red pulp and exhibited a lack of full germinal centers in the white pulp in all the samples, those germinal centers that were present showed up infrequently, had reduced dimensions, and was accompanied by a loss of lymphocytes.

The NC group in Fig 5A has a thymus that is made of a darkly colored cortex that has a high packing of tiny immature lymphocytes. The medulla was seen to have a lighter coloring and less densely cellular composition than the cortex. The thymus demonstrated a decline in the cellular density of cortical lymphocytes, as well as worsening alterations in the remaining epithelial and stromal cells, as evidenced by cytoplasmic vacuolization in Fig 5B, which depicts the DC group. A seemingly typical structure was shown by the thymus in Fig 5C, which included highly populated areas of growing T cells in the thymic cortex and a lesser number of epithelial cells associated with these T cells. In the milk-treated group, the medulla commonly had mature T cells that were bigger in size. Additionally, the medulla contained a greater abundance of epithelial cells and other cell types. The groups of rats to whom the MLMExCC100, MLMExCC200, and MExCC100 were given, the thymus showed a moderate increase in the number of lymphocyte population in both the cortex and the medulla, and macrophages phagocytized apoptotic debris (Fig 5D–5F). Treated with MExCC200 groups exhibited a significant decrease in the number of trabeculae smooth muscle cells and a predominance of collagen fibers.

### 3.2. Anti-inflammatory activity

**3.2.1. Anti-inflammatory effects of *C. chinense* Jacq. in RAW 264.7 cells.** The anti-inflammatory effects of *C. chinense* extracts are presented in Fig 6. The protein expression of both iNOS and COX-2 was found to be inhibited by the *C. chinense* Jacq. extracts. The LPS-pretreated group had significantly higher levels of iNOS and COX-2 expressions than the non-LPS group. However, *C. chinense* extract treatment demonstrated the suppression of iNOS and COX-2 expressions. These findings demonstrate that the plant material has powerful inhibitory effects on the production of NO and early inflammatory factors (Fig 6) and that NO production can be inhibited by inhibiting the expression of iNOS. Additionally, these findings demonstrate that plant material inhibits the production of early inflammatory factors.

**3.2.2. Inhibitory effects of *C. chinense* Jacq. on pro-inflammatory cytokine production.** The inflammatory cytokines TNF-α, IL-6, and IL-1β are all secreted by macrophages after being stimulated by LPS. The possibility that the extracts would have an inhibitory impact on the production of cytokines was investigated (Fig 6). In LPS-stimulated RAW 264.7 cells, the expression of IL-6, and IL-1β was dramatically suppressed in the *C. chinense* Jacq treated group. And the expression of TNF-α was also suppressed. Additionally, these findings demonstrate that plant material inhibits the production of early inflammatory factors.

### 3.3. Effect of *C. chinense* extracts on DNA damage protection

When run through an agarose gel electrophoresis (lane 1), the DNA that was extracted from the pBR322 DNA plasmid appeared as two distinct bands: the band that moved more quickly corresponded to the native form of supercoiled circular DNA (sc DNA), while the band that

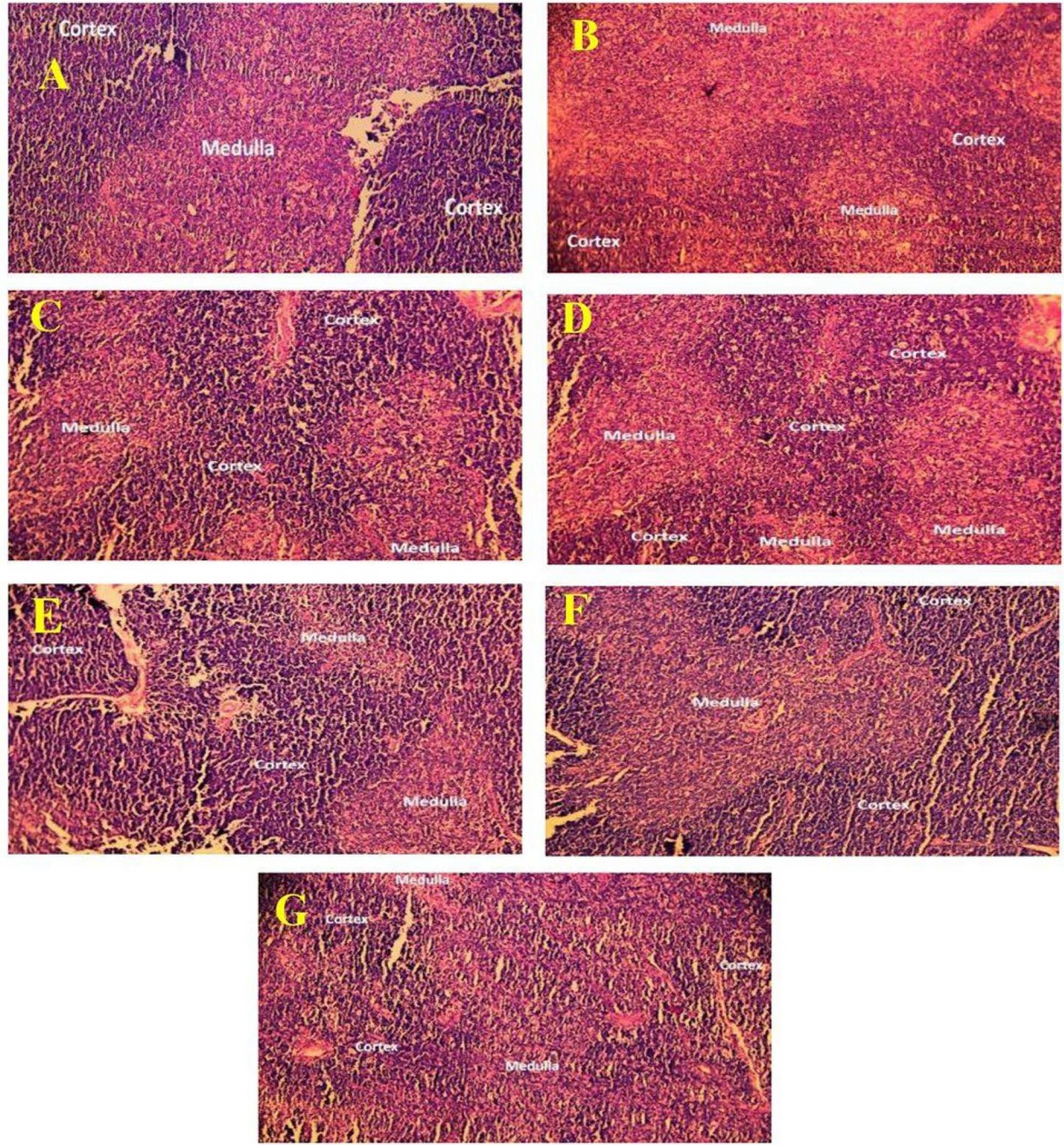

**Fig 5. Image of the thymus tissues from the experimental animal groups as seen through histopathology.** The cortex and medulla of thymus are seen in the image (microscopic resolution: 10 × 40). Hematoxylin and eosin-stained rat thymus micrographs. The images displayed are the glomerulus slices counterstained with hematoxylin and stained with PAS.

moved more slowly was the open circular form of DNA (R- Form). After Fenton's reaction of DNA in the presence of $H_2O_2$, the DNA was cleaved to linear form (L-DNA), which shows that the OH radicals produced by $H_2O_2$ induced DNA strand scission. DNA was cleaved to linear form after Fenton's reaction of DNA in the presence of $H_2O_2$. When compared to

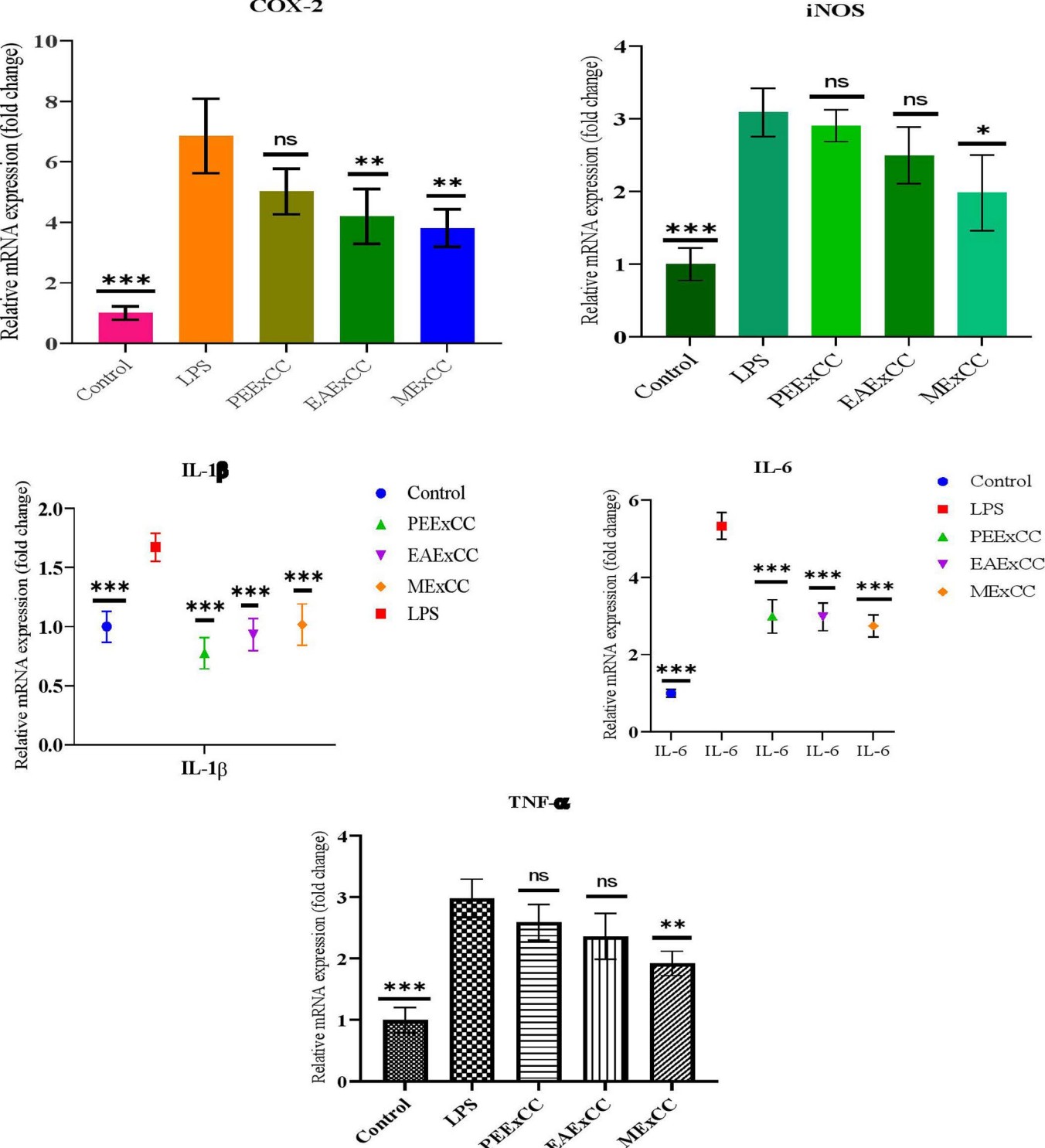

**Fig 6. Effects of PEExCC, EAExCC and MExCC on the expression of pro-inflammatory cytokines in LPS-stimulated RAW 264.7 macrophages.** The expression of (A) cyclooxygenase-2 (COX-2) (B) inducible nitric oxide synthase (iNOS) (C) IL-1β, (D) IL-6 and TNF-α. Values represent the mean ± SD. Statistical significance was calculated by *t*-test and one-way ANOVA. * $p < 0.05$, ** $p < 0.01$, *** $p < 0.001$ indicate significant differences compared to the LPS.

the untreated lane-1, the production of DNA in a repairing form was promoted by adding petroleum ether extract to the reaction mixture at varied concentrations (100, 80, 60, 40, and 20 µg/mL) in lanes 3–7. This resulted in a concentration-dependent action of hydroxyl radical scavenging, which led to the recovery of some of the scDNA. In contrast, there is not even a shred of DNA present in lane 2 of the experiment (Fig 7).

The protection of DNA from damage was carried out by the EAExCC mediated via Fenton's reagent. The electrophoresis gel pattern showed that the varying concentrations of extract (100, 80, 60, 40, and 20 µg/mL) produced distinct patterns (Fig 7). When plasmid DNA was treated in Fenton's reagent without an extract, the reaction produced strand breaking that led to the relaxed form (R- form) (Lane 1–2). However, there was no evidence of recovering native DNA. When incubated with EAExCC, converting native supercoiled DNA did not demonstrate a dose-dependent way nor did it exhibit any dose-dependent effects (Lane 3–7).

In comparison to the untreated lane-1, the formation of L-DNA in a repairing form was stimulated by adding MExCC to the reaction mixture at varying concentrations (100, 80, 60, 40, and 20 µg/mL) in lanes 3–7. This led to partial scDNA recovery due to a concentration-dependent hydroxyl radical scavenging effect. In comparison, there is not even a trace of DNA in lane 2 (Fig 7).

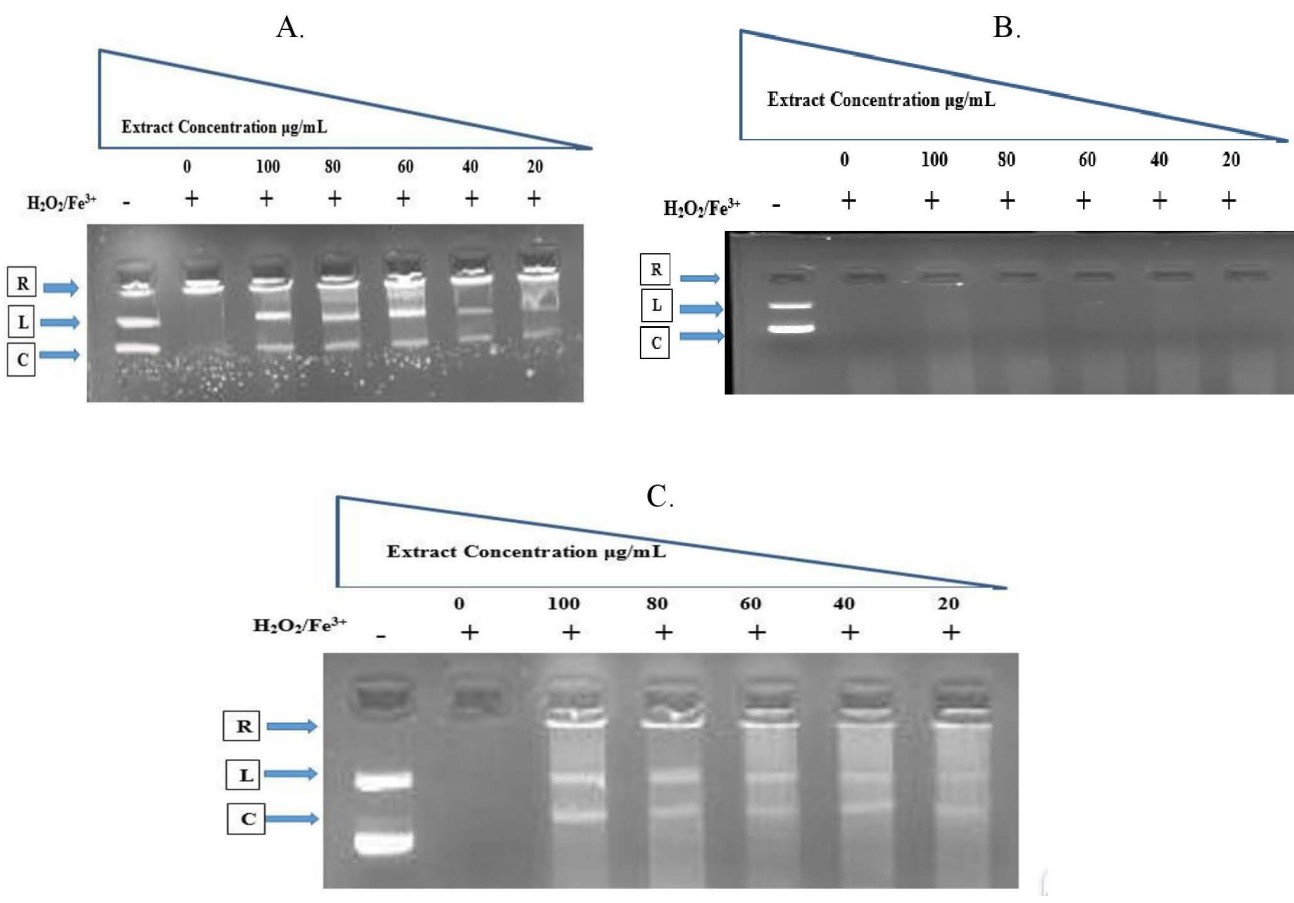

**Fig 7. DNA damage protecting effects of (A) PEExCC; (B) EAExCC and (C) MEExCC.** Electrophoretic pattern of pBR322 supercoiled plasmid DNA after treatment of Fenton's reagent followed by different concentrations. R = single-stranded relaxed nicked DNA (R-Form), L = double-stranded nicked and linear DNA (L-Form), C = native circular DNA (scDNA). For electrophoresis, 1% gel was used with 10 mg/mL ethidium bromide and running in 1x TBE buffer (pH 8.2) at 90 V for 90 min. Lane-1- Untreated plasmid DNA, Lane-2-Treated with Fenton's reagent without extract and Lane 3–7.

### 3.4. Cytotoxic effects of *C. chinense* Jacq. extracts on different cell lines

Extracts of *C. chinense* Jacq. were tested on Vero, BHK-21, and HeLa cell lines to determine their cytotoxic activity. In the presence of PEExCC, EAExCC and MExCC, more than 95% of BHK-21 cells were able to survive (Fig 8A). On the other hand, the survival rate of Vero cells that were treated with PEExCC was < 5%, while the survival rate of same cells treated with EAExCC and MExCC was greater than 95% (Fig 8A). On the HeLa cell line, EAExCC had no discernible effect. Using these extracts, > 95% of the cells were able to survive. However, the survival rate was less than 5% in samples taken from PEExCC and MExCC (Fig 8B).

## 4. Discussion

Natural immunotherapies demonstrate promising alternatives to chemotherapies and vaccines because of their broad range of action, environmentally friendly approaches, and cost effectiveness [24]. After 15 days of consecutive food supplementation in Long Evans, the extracts' effects on the immune system's potentiation were explored in this research. the immunomodulatory activity of *C. chinense* Jacq. combined with milk was examined using the Long Evan rat's model. The use of milk was thought to be due to the fact that *C. chinense* contains capsaicinoids, accounts for pungency, which may have the potential to irritate the rat's stomach. Capsaicinoids irritate the stomach lining and lead to an increase in stomach acid production. Milk protein facilitates emulsification during gastrointestinal digestion. Therefore, milk was used to extensively reduce the pungency of capsaicinoids produced by *C. chinense* [25]. Milk was collected from Bangladesh Standards and Testing Institution (BSTI) recognized standard commercial brand, Arong, which is one of the leading brands in Bangladesh. The collected milk is already standardized by BSTI, and we ensured the standardization. The milk had a composition of fat (3.778 ± 0.018%), Protein (3.282 ± 0.016%), Lactose (4.398 ± 0.045%), and mineral (0.718 ± 0.019). The comprehensive nutritional composition of milk is enclosed in Table 4. This is the first time that this information has been reported. During our research, the extracts had an impact on both the overall body weight of the rats as well as the relative weight of their important organs like the liver. Despite this, there was not a discernible difference between the treatment group and the control group in terms of the body weight of the animals. However, substantial changes were seen between the dosages of 100 and 200 mg/kg body weight in the treated groups. The body weight of the groups that received the highest dose of MExCC200 of tested extracts changed less than that of the groups that acquired the lowest dose of MExCC100 of extracts of the experiment that lasted for 14 days. The body weight continuously decreased as the extract dosage was raised, which suggests that at dosages over 200 mg/kg body weight, the substance is growth-inhibiting rather than growth-enhancing. This is because growth-inhibiting doses are higher than growth-enhancing ones.

The findings are supported by evidence from the liver, a vital organ that responds quickly to any drug. The greatest dosage, 200 mg/kg, resulted in a greater liver mass compared to the lowest dose, which was 100 mg/kg. This indicates that it is hyperactive to offset the impact of the extract. But in the case of other extracts the liver weights were normal, this suggests that the natural activity of the liver was preserved by the other extracts. There was no difference in the weight of the spleen between the monitoring group and the recovery group, but there was a significant difference in the weight of the thymus between the other extract groups and the high-dose extract group. The researchers Bin-Hafeez et al. [15] and Sumalatha et al. [26] evaluated the immunomodulatory effectiveness of fenugreek and *Salacia chinensis* extracts in mice. They found that there was no substantial change in the animals' body weight, and there was also no influence on the size of the spleen.

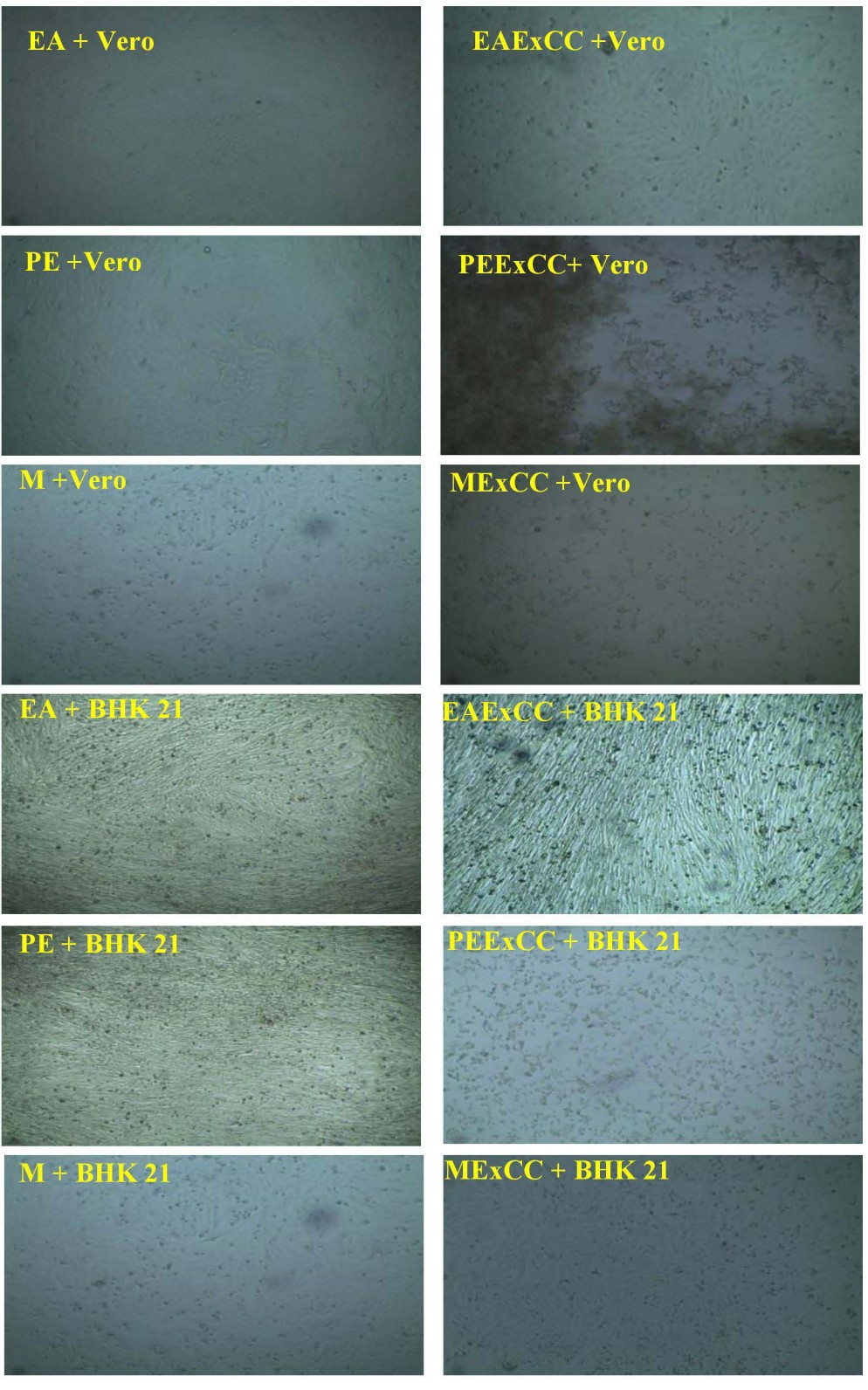

A.

**Fig 8. (A) Pictoral view of cytotoxic activity of EAExCC, PEExCC and MExCC in Vero and BHK 21 cells; (B) Pictoral view of cytotoxic activity of EAExCC, PEExCC and MExCC in HeLa cells.**

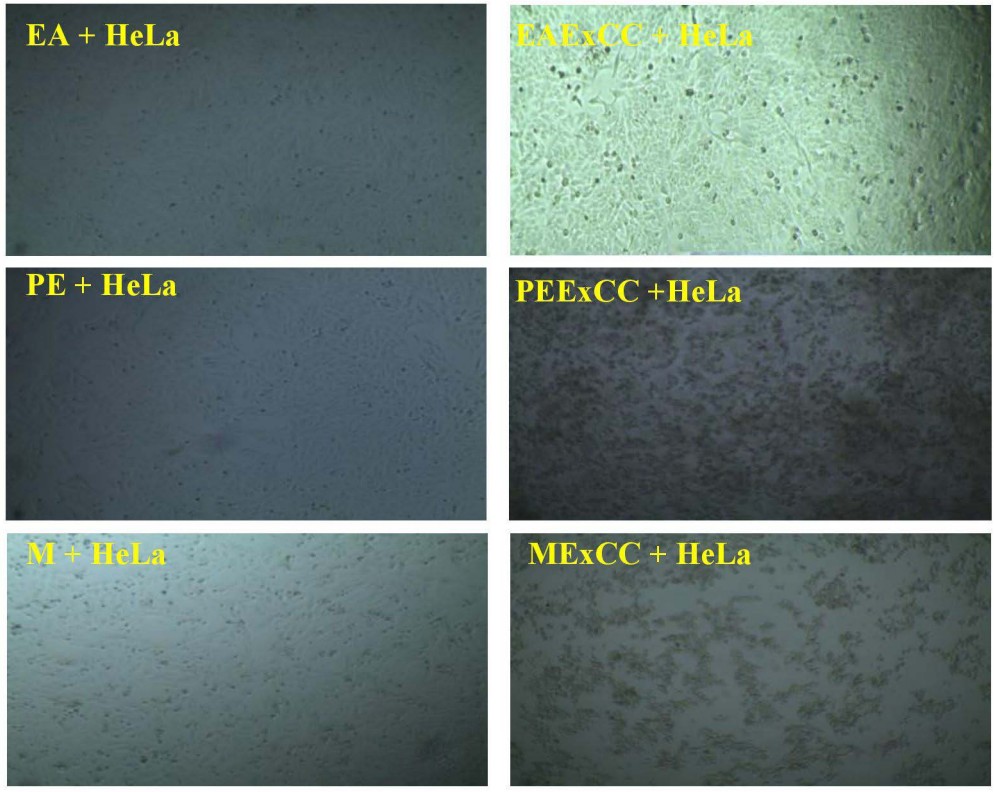

**B.**

Fig 8. Continued.

Table 4. The composition of milk used with the treatment.

| Milk constituent | Percentage (%) of the constituent |
|---|---|
| Fat percentage | 3.778 ± 0.018 |
| Protein percentage | 3.282 ± 0.016 |
| Lactose percentage | 4.398 ± 0.045 |
| SNF percentage | 8.345 ± 0.028 |
| Density percentage | 1.024 ± 0.001 |
| Mineral percentage | 0.718 ± 0.019 |
| Freezing point | 0.4773 ± 0.004 |
| Specific gravity | 1.029 ± 0.0060 |

The animal's hematological characteristics show and decide one of the initial immune responses to a pathogen. The total leukocyte counts, as well as the differential leukocyte count, were assessed, as was the control group as well as the pretreatment groups with varying extract dosages. In this investigation, we investigated the immunological components and mediators that were present in rats that had been given extracts of *C. chinense* Jacq. that had been subjected to a challenge. After being persuasively fed on each of the several extracts, the rats' hematologic parameters were found to have undergone a considerable shift, as shown by the findings of our study. At dosages of ML, MLMExCC100 of the *C. chinense* Jacq. extracts, there was a considerable rise in the WBCs. MLMExCC200 and MExCC100 extract exhibited a considerable rise in the WBCs compared to the untreated and negative control groups,

with a modest increase from the positive control group. On the other hand, the WBCs count declined in the highest dose feeding extract group.

Also, neutrophils are part of the cell-mediated immune responses that are responsible for the innate immunity that contributes to the clearance of foreign bodies by recognizing and migrating toward the foreign body, engaging in phagocytosis, and destroying the foreign agent [27]. The presence of a wide variety of chemicals, macronutrients, and micronutrients in herbs may be responsible for the higher percentage of neutrophil adhesion [28]. This makes a significant contribution to the capacity of neutrophils to migrate in the direction of an infectious pathogen. Because of their crucial function in calcium and bone metabolism, micronutrients like vitamin D are required for the formation of polymorphonuclear neutrophils. This is because these micronutrients help maintain healthy bone tissue. In the current study, the groups ML, MLMExCC200, MLMExCC100, and MExCC100, had the maximum neutrophil count of 59.2%, 59.2%, 60.8%, 60% WBC/mm³ indicating the blood cells were being triggered to launch a potent immune response. The results showed that the other extracts elicit a stronger immune response than MExCC200. Since the plant extract increases the phagocytic activity of the neutrophils, it might be employed to boost the cell-mediated immune response in cases when the immune system is already impaired.

According to the categorization of hypersensitivity responses developed by Coombs and Gell in 1975 [29], the DTH reaction is a cell-mediated immune response of the type IV kind. The increase in footpad thickness of the rats that were subjected to *C. chinense* extracts in this study could be attributed to the ability of the extract to activate lymphocytes and their accessory cell types, leading to an enhancement in the production of antibodies in animals that had previously been immunosuppressed, thereby increasing cell-mediated immunity. The rats were given *C. chinense* Jacq. extracts via intraperitoneal injection. Vitamins A and C have been identified as being present in *C. chinense* Jacq [30]. These substances activate the immune system by promoting T-cell proliferation, increasing cytokine release, and synthesis of immunoglobulins [31]. All these factors play an essential role in the inflammatory response, which was seen as an increase in the thickness of the footpad. In addition, the extract of *C. chinense* Jacq. contains amino acids that play a crucial role in the production of immunoglobulins and major histocompatibility complexes, both of which are required for the successful mediation of the DTH response. Trace elements are necessary for the proliferation of T-cells and Langerhans cells as well as the activity of lytic enzymes, both of which are crucial components of the DTH response to antigen. Trace elements may be found in a variety of plant and animal foods. The delayed type of hypersensitivity reaction is one of the parameters that is often used to evaluate an animal's cell-mediated immune response. Evaluation of the immunomodulatory activities of *Momordica charantia ghrita* extract carried out on albino rats was carried out by Prasad et al. [32] using a DTH experiment. When the medicine was given at a dosage of 350 mg/kg per day, scientists found that there was a significant increase in the volume of the paws. In the instance of the rats that were given herbal medications, Pradhan et al. [33] demonstrated that there was an enhanced hypersensitive response. In the same vein as earlier findings, Bin-Hafeez et al. [15] demonstrated that when fenugreek was utilized at a concentration of 50 mg/kg, a significant increase in delayed type hypersensitivity response was observed in comparison to the control. This was the case when looking at the results of the experiment. In addition, Fulzel et al. [34] demonstrated that *Ashtamangala ghrita* has the ability to shed light on an elevated delayed type hypersensitivity reaction. In line with the findings described in the previous papers, our high-dose extracts induced a more delayed type of hypersensitivity reaction. Since there was a dose-dependent increase in paw size in response to antigen, the findings of the research demonstrated that the MExCC may be utilized to stimulate the immune system.

The ratio of an individual's serum protein to their serum albumin globulin is one of the early markers that an individual's blood chemistry is normal. If there was a shift in the serum protein content as well as the albumin ratio, this would provide us with a clue about the individual's changed immune response state. According to the findings of the current study, the serum protein level and albumin globulin ratio are found to be lower in the case of higher concentrations of MExCC200 of the drug. However, in the other concentrations of extracts test, the group showed an increase in serum protein and albumin ratio, indicating that higher immune responses might have contributed to the serum protein in terms of a variety of different molecules such as immunoglobulins and other humoral factors.

Several anti-inflammatory bioactivities in plants have recently been linked through epidemiological research. As a result, conventional pharmaceuticals or natural products may be a source of new anti-inflammatory active molecules, eventually leading to novel therapies with less hazardous side effects [35,36]. By preventing cell damage and death from inflammation and apoptosis, antioxidants play a crucial role in redox pathways [37]. Overactivation of macrophages and subsequent excessive inflammation may result from oxidative stress brought on by LPS or other triggers [38]. Here, we provided strong evidence that the extracts inhibited LPS-induced generation of TNF-α, intracellular ROS in RAW 264.7 macrophages. This provides evidence that the extracts have anti-inflammatory qualities that may help with a variety of inflammatory symptoms. Inhibition of NF-kB and MAPK pathway activation in LPS-stimulated RAW 264.7 cells has been shown before to reduce iNOS, COX-2, TNF-α, and IL-6 production [39]. We found evidence that the extracts had potent antioxidant effects, maybe because of the large quantities of polyphenolic chemicals they contain. Based on the results of previous research [40], it was determined that LPS induces oxidative stress, which in turn may activate macrophages and result in an inflammatory response that is too strong. So, we dug further into the antioxidant actions that lay behind the reduction in LPS-induced macrophage activation. In macrophages, the production of ROS is quickly boosted in response to LPS stimulation. Furthermore, this reaction induces DNA damage and results in an increase in inflammatory cytokines and enzymes [41]. During inflammation, the immune system produces cytokines (TNF-α, IL-6, IL-1β) and enzymes (iNOS, COX-2) that promote further inflammation. To our surprise, our extract suppressed the production of these cytokines and enzymes. Local inflammation and sepsis are primarily mediated by IL-6 and TNF-α [42]. Increased IL-1β expression was seen in several human malignancies, including melanoma, colon, breast, and lung tumors [43]. This indicates that the extracts not only dampen the inflammatory response but also stimulate the production of anti-inflammatory molecules. It is imperative that the precise mechanism(s) be responsible for *C. chinense* Jacq. anti-inflammatory properties be determined so that medicines may be designed to treat inflammatory illnesses.

It is well known that OH-free radicals cause damage to cellular DNA. Most of the oxidative damage that occurs to proteins and DNA is caused by OH radicals, which are produced when $H_2O_2$ is photolyzed by UV light. According to Guha et al. [44], when OH radicals attach to DNA, this results in strand breaking and opening, as well as fragmentation of deoxyribose sugar and alteration of nitrogenous bases. In most cases, OH radicals are to blame for the oxidative damage that is done to DNA, which is one of the most critical processes in the beginning stages of the development of cancer [45]. All the extracts were tested to see whether they could preserve a model DNA called pBR322 from oxidative damage. It is well established that the hydroxyl radicals produced by Fenton's reaction may damage DNA, as shown by the absence of the DNA band in lanes 2. Even though all the extracts, except for EXExCC, effectively mitigated the oxidative stress and protected the DNA from the hydroxyl radicals generated by Fenton's reaction, as confirmed by the presence of DNA bands (Fig 10) appears to be comparably most effective in maintaining the DNA's intact state respectively. DNA

strands may be broken and damaged by free radicals, which can ultimately lead to carcinogenesis, mutagenesis, and cytotoxicity. Free radicals are notorious for their harmful effects [46,47]. Several groups of researchers have come to the same conclusions and employed plant extracts and fractions to protect DNA from oxidative damage [48]. In this study, the above two extracts displayed considerable protective activity against free radical-mediated DNA damage and could, therefore, be used in cancer prevention.

Furthermore, the BHK-21 and Vero cell lines were not too affected by the EAExCC and MExCC showing that they are sustainable. PEExCC, on the other hand, was harmful to Vero cell lines. PEExCC, and MExCC were shown to reduce the growth of the cancer cell line HeLa. The effects of *C. chinense* Jacq. extracts on BHK21, vero, and cervical cancer cell lines are compared for the first time in this work. Three extracts of *C. chinense* Jacq. were tested for their cytotoxic effects on the HeLa cell line in the present investigation. The HeLa cell line was found to be sensitive to PEExCC and MExCC cytotoxic effects. Both extracts were more effective against the HeLa cell line than the EAExCC. When tested on three different cell lines, methanol extract showed the most promising. Plant phytochemicals [49] trigger apoptosis through the intrinsic route by inducing the release of cytochrome-c in certain tumor cell types. Tomatoes, which contain the carotenoid phytochemical lycopene [50], chilli peppers, which contain the phenolic compound capsaicin [51], and celery, which contain the flavonoids luteolin and curcumin [52], all of which depolarize mitochondria and induce apoptosis in human tumor cells [53].

## 5. Conclusion

When compared to the capsaicin levels found in the fruits of other chilli species, it has been shown that the levels of capsaicin are found in *C. chinense* Jacq. fruits are very high. Analgesic, antiarthritic, anticancer, and antioxidant activities are some of the attributes of *C. chinense* Jacq. that make it beneficial in the field of pharmacology. Only just a handful of research papers on *C. chinense* Jacq. have been published up to this point, even though it has the potential to be a source of antioxidants in addition to possessing other pharmacological properties. According to the findings of this study, the extracts may safeguard DNA from the damaging effects of oxidative stress. The outcomes of this investigation showed, overall, that the immune response is greatly improved when the methanolic extracts are given as a treatment. According to the results, drug extracts taken in lower amounts may activate both a humoral and cellular immune response in the body. It has been shown that higher dosages, or doses that are greater overall hinder the immune system's capacity to mount a response. In conclusion, the extracts improve the body's immune system. However, it is important to note that a high concentration of the extracts may have a biochemical impact on the body. Across the studied biological functions of *C. chinense* Jacq, it is anticipated that *C. chinense* itself or its biometabolites could serve as vibrant therapeutic sources for alleviating immune complications. To comprehend the underlying mechanisms and safety levels, comprehensive studies on bioavailability, preclinical pharmacokinetics, and toxicity are essential before advancing to clinical trials for effective and safe immunomodulatory development agents.

## Supporting information

**S1 Fig. DNA damage protecting effects of PEExCC.**
(JPG)

**S2 Fig. DNA damage protecting effects of EAExCC.**
(JPG)

**S3 Fig. DNA damage protecting effects of MEExCC.**
(JPG)

## Acknowledgment

The authors wish to acknowledge the members of the Laboratory of alternative Medicine and Natural Products Research for their undeniable support to accomplish the research.

## Author contributions

**Conceptualization:** Md. Atiar Rahman.

**Data curation:** Srabonti Saha, Humayra Ferdousi, Md. Mannan, Md. Asif Nadim Khan.

**Formal analysis:** Srabonti Saha, Jobaier Ibne Deen, Md. Mannan.

**Funding acquisition:** Md. Atiar Rahman.

**Investigation:** Srabonti Saha, Md. Altaf Hossain, Humayra Ferdousi, Akhlak Chowdhury, Md. Mannan.

**Methodology:** Srabonti Saha, Md. Altaf Hossain, Humayra Ferdousi, Akhlak Chowdhury, Md. Mannan.

**Project administration:** Md. Atiar Rahman.

**Resources:** Akhlak Chowdhury, Md. Asif Nadim Khan.

**Software:** Jobaier Ibne Deen, Md. Asif Nadim Khan.

**Supervision:** Md. Atiar Rahman.

**Visualization:** Jobaier Ibne Deen.

**Writing – original draft:** Srabonti Saha.

**Writing – review & editing:** Md. Atiar Rahman.

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
