## [Decision Letter · Decision Letter 0]

13 Sep 2024

PONE-D-24-29500Capsicum chinense Jacq. fruit plays an immunomodulatory roles cytokine attenuation and DNA damage protectionPLOS ONE

Dear Dr. Rahman,

Thank you for submitting your manuscript to PLOS ONE. After careful consideration, we feel that it has merit but does not fully meet PLOS ONE’s publication criteria as it currently stands. Therefore, we invite you to submit a revised version of the manuscript that addresses the points raised during the review process.

**ACADEMIC EDITOR**Although I find this work intriguing, I would like the authors address some concerns.

The collection of fruit and verification of its authenticity gives rise to certain concerns. The fruit was obtained from the farmers market. By what means was the fruit determined to be free from any adulteration?.  Based on what criteria was the botanical number assigned?

The authors of the experimental design included the use of milk in some groups. What was the rationale for using milk? Please specify the origin and nutritional composition of the milk.

Please explicitly outline the immunisation procedures with relevance to the research design.

Could you kindly specify the in-vivo and in-vitro research separately?  Please submit your revised manuscript by Oct 28 2024 11:59PM. If you will need more time than this to complete your revisions, please reply to this message or contact the journal office at plosone@plos.org . Please include the following items when submitting your revised manuscript:

We look forward to receiving your revised manuscript.

Kind regards,

Sairah Hafeez Kamran, PhD

Academic Editor

PLOS ONE

**Journal Requirements:**

2. To comply with PLOS ONE submissions requirements, in your Methods section, please provide additional information regarding the experiments involving animals and ensure you have included details on (a) methods of sacrifice, (b) methods of anesthesia and/or analgesia, and (c) efforts to alleviate suffering.

3. In the online submission form, you indicated that Data will be available upon request.

Reviewers' comments:

Reviewer's Responses to Questions

**Comments to the Author**

1. Is the manuscript technically sound, and do the data support the conclusions?

Reviewer #1: Yes

Reviewer #2: Partly

2. Has the statistical analysis been performed appropriately and rigorously? 

Reviewer #1: Yes

Reviewer #2: Yes

3. Have the authors made all data underlying the findings in their manuscript fully available?

Reviewer #1: Yes

Reviewer #2: Yes

4. Is the manuscript presented in an intelligible fashion and written in standard English?

Reviewer #1: Yes

Reviewer #2: No

5. Review Comments to the Author

**Reviewer #1:**  The manuscript is well written, however according to my review minor changes are required which are as follows:

Page 4 of Abstract in line no. 3, the statement can be rephrased as Immunomodulatory effect of CC was studied in SRBC-induced animal models along with the evaluation of anti inflammatory effects of LPS-stimulated RAW264.7 macrophages.

Page 6 of introduction in line no. 3, line should be rephrased as extensive study for the development of new methods needs to be done to create innovative immunomodulators from plants. In line 13, pharmacological importance of CC necessitates the investigation of immunomodulatory and relevant biological functions of CC and its products.

Page 7, Material and methods, in line 18, the sentence should be, resulting powder was defatted using n-hexane and was extracted using petroleum ether, ethyl acetate and methanol for 3days respectively.

Page 8, Experimental design, line 6 should include acclimatization of animals before grouping details.

Page 9, immunization, dosing schedule needs to be simplified and extraction of sheep blood cells needs to be elaborated.

Page 9, body weight and lymphoid organ, line 15 blood was collected using cardiac puncture using 21G needle and then animals were anesthetized.

Page 11 line 20, morphological study after incubation of 72 hours morphological changes were studied under light microscope.

Page 12 preparation of reagents needs to be simplified.

Page 13 DNA damage protection line 13, is not making sense some line/word is missing. line 22 reaction mixture is allowed to stand for an hour (word is missing)

Discussion page 21, In line 5, Rephrased sentence is natural immunotherapies demonstrate promising alternatives to chemotherapies and vaccines. In line 15 the body weight of the groups that received highest dose.

Page 22 line 9, the animals hematological characteristics are indicative of initial immune response to pathogens.

**Reviewer #2:**  Amazing work has been done. The presentation of work, however, needs improvement and revision. The English needs improvement. Likewise, the presentation of results and explanation of procedures need to be revised.

6. PLOS authors have the option to publish the peer review history of their article (what does this mean? ). If published, this will include your full peer review and any attached files.

**Do you want your identity to be public for this peer review?** For information about this choice, including consent withdrawal, please see our Privacy Policy .

Reviewer #1: No

Reviewer #2: No

---

## [Author Response · Author response to Decision Letter 1]

10 Oct 2024

We note that you have included affiliation numbers 1 and 2 however only affiliations 1 have authors linked to them. Please amend affiliation 2 to link an author to it or remove if added in error.

Author’s Response: Thank you so much for bringing the issue to the attention of the authors. Authors have addressed the issue and removed the affiliation 2.

---

## [Decision Letter · Decision Letter 1]

1 Nov 2024

PONE-D-24-29500R1Capsicum chinense Jacq. fruit plays an immunomodulatory role in a cytokine attenuation and DNA damage protectionPLOS ONE

Dear Dr. Rahman,

Thank you for submitting your manuscript to PLOS ONE. After careful consideration, we feel that it has merit but does not fully meet PLOS ONE’s publication criteria as it currently stands. Therefore, we invite you to submit a revised version of the manuscript that addresses the points raised during the review process.

**ACADEMIC EDITOR:**
**The authors did not address several major points in the revision. **

**The collection of fruit and verification of its authenticity gives rise to certain concerns. The fruit was obtained from the farmers market. By what means was the fruit determined to be free from any adulteration?.  Based on what criteria was the botanical number assigned?**

**The experimental design included the use of milk in some groups. What was the rationale for using milk? Please specify the origin and nutritional composition of the milk. Please specify that you have standardized the milk composition.**

**Please specify the in-vitro and in-vivo procedures separately?. **

We look forward to receiving your revised manuscript.

Kind regards,

Sairah Hafeez Kamran, PhD

Academic Editor

PLOS ONE

Reviewers' comments:

Reviewer's Responses to Questions

**Comments to the Author**

1. If the authors have adequately addressed your comments raised in a previous round of review and you feel that this manuscript is now acceptable for publication, you may indicate that here to bypass the “Comments to the Author” section, enter your conflict of interest statement in the “Confidential to Editor” section, and submit your "Accept" recommendation.

Reviewer #2: All comments have been addressed

2. Is the manuscript technically sound, and do the data support the conclusions?

Reviewer #2: Yes

3. Has the statistical analysis been performed appropriately and rigorously? 

Reviewer #2: Yes

4. Have the authors made all data underlying the findings in their manuscript fully available?

Reviewer #2: Yes

5. Is the manuscript presented in an intelligible fashion and written in standard English?

Reviewer #2: Yes

6. Review Comments to the Author

Reviewer #2: Author made the changes needed and have submitted them. For future papers author can improve writing style and expression. In my opinion, this paper should be given a chance to get published.

7. PLOS authors have the option to publish the peer review history of their article (what does this mean? ). If published, this will include your full peer review and any attached files.

**Do you want your identity to be public for this peer review?** For information about this choice, including consent withdrawal, please see our Privacy Policy .

Reviewer #2: No

---

## [Author Response · Author response to Decision Letter 2]

12 Nov 2024

Dear Editor

Thank you so much for your time and patience to leave the comments on our revised version of the manuscript entitled “Capsicum chinense Jacq. fruit plays an immunomodulatory role in a cytokine attenuation and DNA damage protection” to your esteemed journal “Plos One”. We have gone through the comments your comments and responded in a point-to-point manner as follows:

The collection of fruit and verification of its authenticity gives rise to certain concerns. The fruit was obtained from the farmers market. By what means was the fruit determined to be free from any adulteration?

Dear editor,

Thank you so much to repeat the questions. Let us answer the questions’ part by part”

Editor’s query: The collection of fruit and verification of its authenticity gives rise to certain concerns. The fruit was obtained from the farmer’s market.

Author’s response: The fruits were collected from the farmers market, however; the quality of fruits was ensured to be free from adulteration. The major criteria were the colour, appearance, texture, flavour, weight, and volume. No spotted/rotten/unhealthy/infectious fruits were accepted. Very fresh and natural fruits were accepted for our research.

Based on what criteria was the botanical number assigned?

The fruit was identified and botanically assigned by Dr. Shaikh Bokhtear Uddin (bokhtear@cu.ac.bd) who is a plant taxonomist and Professor of the department of Botany, University of Chittagong. However, the following approaches were maintained to prepare and taxonomically identify the sample.

Plant description was made including the plant habit (tree, shrub, vine, herb), height, growth form, color, bark, branching, leaf orientation, or general volume/size/spread of a plant. Detailed location of the plant, origin of material, habitat, frequency (whether the plant is rare or common), herbarium sheet preparation (pressing and drying the sample, labelling, mounting), comparing with the published plant descriptions, illustrations and photographs to identify the herbarium specimens.

The experimental design included the use of milk in some groups. What was the rationale for using milk?

The capsaicinoids of C. chinense generally accounts for their pungency. Capsaicinoids irritate the stomach lining and lead to an increase in stomach acid production. Milk protein facilitates an emulsification during gastrointestinal digestion. Therefore, milk was used to extensively reduce the pungency of capsaicinoids produced by C. chinense.

Please specify the origin and nutritional composition of the milk. Please specify that you have standardized the milk composition.

Milk was collected from Bangladesh Standards and Testing Institution (BSTI) recognized standard commercial brand, Arong, which is one of the leading brands in Bangladesh. The collected milk is already standardized by BSTI, and we ensured the standardization. The nutritional composition of milk is as follows:

Fat percentage 3.778 ± 0.018

Protein percentage 3.282 ± 0.016

Lactose percentage 4.398 ± 0.045

SNF percentage 8.345 ± 0.028

Density percentage 1.024 ± 0.001

Mineral percentage 0.718 ± 0.019

Freezing point 0.4773 ± 0.004

Specific gravity 1.029 ± 0.0060

Please specify the in-vitro and in-vivo procedures separately?

Thank you so much for your suggestion. In fact, the in vivo and ain vitro procedures are presented separately.

The In vivo Immunomodulating activity of C. chinense Jacq, with Long Evans rats and follow-up experimentations (immunization, Body weight and lymphoid organ weight, total serum protein and albumin: globulin ratio, total leukocyte count, differential count etc) whereas in vitro experimentations (Anti-Inflammatory activity against LPS-stimulated RAW264.7 macrophages) were carried out with follow-up experimentation.

---

## [Editor Report · Decision Letter 2]

15 Nov 2024

PONE-D-24-29500R2Capsicum chinense Jacq. fruit plays an immunomodulatory role in a cytokine attenuation and DNA damage protectionPLOS ONE

Dear Dr. Rahman,

Thank you for submitting your manuscript to PLOS ONE. After careful consideration, we feel that it has merit but does not fully meet PLOS ONE’s publication criteria as it currently stands. Therefore, we invite you to submit a revised version of the manuscript that addresses the points raised during the review process. 

We look forward to receiving your revised manuscript.

Kind regards,

Sairah Hafeez Kamran, PhD

Academic Editor

PLOS ONE

Journal Requirements:

**Additional Editor Comments:**

The authors are requested to include the standardization procedure of the milk in the manuscript.

You are also requested to add the season and year of collection of Capsicum chinense Jacq as well as precise latitude and longitude of the collection site.

---

## [Author Response · Author response to Decision Letter 3]

20 Nov 2024

Response to reviewers

Dear Editor

Thank you so much for your time and patience to leave the comments on our revised version of the manuscript entitled “Capsicum chinense Jacq. fruit plays an immunomodulatory role in a cytokine attenuation and DNA damage protection” to your esteemed journal “Plos One”. We have gone through the comments your comments and responded in a point-to-point manner as follows:

The collection of fruit and verification of its authenticity gives rise to certain concerns. The fruit was obtained from the farmers market. By what means was the fruit determined to be free from any adulteration?

Dear editor,

Thank you so much to repeat the questions. Let us answer the questions’ part by part”

Editor’s query: The collection of fruit and verification of its authenticity gives rise to certain concerns. The fruit was obtained from the farmer’s market.

Author’s response: The fruits were collected from the farmers market, however; the quality of fruits was ensured to be free from adulteration. The major criteria were the colour, appearance, texture, flavour, weight, and volume. No spotted/rotten/unhealthy/infectious fruits were accepted. Very fresh and natural fruits were accepted for our research.

Based on what criteria was the botanical number assigned?

The fruit was identified and botanically assigned by Dr. Shaikh Bokhtear Uddin (bokhtear@cu.ac.bd) who is a plant taxonomist and Professor of the department of Botany, University of Chittagong. However, the following approaches were maintained to prepare and taxonomically identify the sample.

Plant description was made including the plant habit (tree, shrub, vine, herb), height, growth form, color, bark, branching, leaf orientation, or general volume/size/spread of a plant. Detailed location of the plant, origin of material, habitat, frequency (whether the plant is rare or common), herbarium sheet preparation (pressing and drying the sample, labelling, mounting), comparing with the published plant descriptions, illustrations and photographs to identify the herbarium specimens.

The experimental design included the use of milk in some groups. What was the rationale for using milk?

The capsaicinoids of C. chinense generally accounts for their pungency. Capsaicinoids irritate the stomach lining and lead to an increase in stomach acid production. Milk protein facilitates an emulsification during gastrointestinal digestion. Therefore, milk was used to extensively reduce the pungency of capsaicinoids produced by C. chinense.

Please specify the origin and nutritional composition of the milk. Please specify that you have standardized the milk composition.

Milk was collected from Bangladesh Standards and Testing Institution (BSTI) recognized standard commercial brand, Arong, which is one of the leading brands in Bangladesh. The collected milk is already standardized by BSTI, and we ensured the standardization. The nutritional composition of milk is as follows:

Fat percentage 3.778 ± 0.018

Protein percentage 3.282 ± 0.016

Lactose percentage 4.398 ± 0.045

SNF percentage 8.345 ± 0.028

Density percentage 1.024 ± 0.001

Mineral percentage 0.718 ± 0.019

Freezing point 0.4773 ± 0.004

Specific gravity 1.029 ± 0.0060

Please specify the in-vitro and in-vivo procedures separately?

Thank you so much for your suggestion. In fact, the in vivo and ain vitro procedures are presented separately.

The In vivo Immunomodulating activity of C. chinense Jacq, with Long Evans rats and follow-up experimentations (immunization, Body weight and lymphoid organ weight, total serum protein and albumin: globulin ratio, total leukocyte count, differential count etc) whereas in vitro experimentations (Anti-Inflammatory activity against LPS-stimulated RAW264.7 macrophages) were carried out with follow-up experimentation.

---

## [Decision Letter · Decision Letter 3]

22 Dec 2024

PONE-D-24-29500R3Capsicum chinense Jacq. fruit plays an immunomodulatory role in a cytokine attenuation and DNA damage protectionPLOS ONE

Dear Dr. Rahman,

Thank you for submitting your manuscript to PLOS ONE. After careful consideration, we feel that it has merit but does not fully meet PLOS ONE’s publication criteria as it currently stands. Therefore, we invite you to submit a revised version of the manuscript that addresses the points raised during the review process.

ACADEMIC EDITOR: In my earlier remarks, I stated that writers should include the composition of milk in the manuscript text; however, this was not included in the submitted revision. />==============================

**A rebuttal letter that responds to each point raised by the academic editor and reviewer(s). You should upload this letter as a separate file labeled 'Response to Reviewers'.****A marked-up copy of your manuscript that highlights changes made to the original version. You should upload this as a separate file labeled 'Revised Manuscript with Track Changes'.****An unmarked version of your revised paper without tracked changes. You should upload this as a separate file labeled 'Manuscript'.**

We look forward to receiving your revised manuscript.

**Kind regards,**

Sairah Hafeez Kamran, PhD

Academic Editor

PLOS ONE

Journal Requirements:

Reviewers' comments:

Reviewer's Responses to Questions

**Comments to the Author**

1. If the authors have adequately addressed your comments raised in a previous round of review and you feel that this manuscript is now acceptable for publication, you may indicate that here to bypass the “Comments to the Author” section, enter your conflict of interest statement in the “Confidential to Editor” section, and submit your "Accept" recommendation.

**Reviewer #3: (No Response)**

**Reviewer #4: All comments have been addressed**

**Reviewer #5: (No Response)**

2. Is the manuscript technically sound, and do the data support the conclusions?

**Reviewer #3: Yes**

**Reviewer #4: Yes**

**Reviewer #5: Yes**

**3. Has the statistical analysis been performed appropriately and rigorously? **

**Reviewer #3: Yes**

**Reviewer #4: Yes**

**Reviewer #5: Yes**

4. Have the authors made all data underlying the findings in their manuscript fully available?

**Reviewer #3: Yes**

**Reviewer #4: Yes**

**Reviewer #5: Yes**

5. Is the manuscript presented in an intelligible fashion and written in standard English?

**Reviewer #3: Yes**

**Reviewer #4: Yes**

**Reviewer #5: Yes**

6. Review Comments to the Author

Reviewer #3: Review Report:

The manuscript titled "Capsicum chinense Jacq. fruit plays an immunomodulatory role in a cytokine attenuation and DNA damage protection" suggests that Capsicum chinense Jacq. fruit has potential health benefits by modulating the immune system. It implies the fruit can reduce the overproduction of cytokines, which are signaling proteins involved in inflammation. Additionally, the fruit may protect against DNA damage, highlighting its role in cellular protection and overall immune health. Overall, the manuscript appears to present a promising study with implications for both basic research and clinical practice which highlights its potential as a natural therapeutic agent for enhancing immune health and cellular resilience.

Note: I reviewed the file named (PONE-D-24-29500-R3) downloaded from journal online portal and all my comments are according to the line numberings of the pdf file “PONE-D-24-29500-R3”

Detailed Comments/Recommendations:

Title:

The title of the study, "Capsicum chinense Jacq. fruit plays an immunomodulatory role in a cytokine attenuation and DNA damage protection" appears to be apt and engaging. It effectively highlights the fruit's potential therapeutic significance, appealing to researchers and healthcare professionals interested in natural interventions for inflammation and cellular protection.

Abstract:

It will be more attractive for the readers if the authors may add up aims and objective of the study at the start of abstract in 2 lines.

Page 4, line 3: Mention the animal (rats).

Page 4, Line 6: Mention the dose at which immune response was suppressed.

It will be more appropriate if the authors may add few numerical results in abstract. It will be more impressive for readers.

Key Words: Well written.

Introduction:

Page 5, Lines 8-13: Sentence is too long; it will be more impressive if authors may split into portions for the better understandings of the reader.

Page 5, Line 15-18: The statement is confusing for the reader. It can be rewritten for the ease of reader.

Page 6, Lines 4 to onward: Please add biography of Capsicum chinense Jacq, along with its habitat and cultivations etc.

Materials and Methods:

Page 7, Line 8: How much Capsicum chinense Jacq. Was obtained? Please mention its quantity.

Page 8, Line 6: Please mention temperature and pressure adjusted in rotary evaporator.

Page 8, Line 18: Kindly mention humidity conditions, provided to animals in animal house.

Page 8, Line 3: Please mention quantity of antigen administered to animals along with route of administration.

Page 10, Line 3: Please insert reference for the methodology of immunization.

Pages 10-11: References are missing for all the methods mentioned on page 10 and 11.

Page 11, Line 20: Please mention magnification/resolution power at which slides were observed under microscope.

Page 15, Line 4: please insert reference.



Results:

Page 16, Line 12-14: Please write the sentence in past tense.

All results are well expressed

Discussion:

Page 21, Line 17-18: give full form of bw.

Over all the discussion is well written and quite satisfactory.

Conclusion:

It is well written. The authors are requested to add on future prospects of the study in the last of conclusion.

References:

Page 33, Line 7: The page numbers are missing.

The authors are advised to review all the references for the strict adherence of the uniformity of Journal’s references style.

Tables:

Table 2: Please add unit (g) in the column of weight change.

Concluding Remarks:

The paper necessitates the aforementioned corrections, inclusive of rectifying any errors highlighted, coupled with a meticulous proofreading to refine its English language usage. Upon addressing these amendments, the paper will meet the required standards for acceptance.

Reviewer #4: The manuscript entitles as “Capsicum chinense Jacq. fruit plays an immunomodulatory role in a cytokine attenuation” is written by Saha et al., is on the role of Capsicum chinense’s fruits in cytokine attenuation. The manuscript is interesting and can be a new addition to the literature but needs to address some comments before acceptance.

1.The graphic abstract needs to be revised by adding the necessary images only to make it more concise and informative.

2.Did the authors isolate/identified chemical compounds from the study plant’s fruits and screen them for these biological activities. The activity of the extracts can be due to the synergistic effect of the mixture of compounds and pure compounds can be more prominent and effective.

3.Why did the authors select methanol extract and use two doses of high concentrations (100 and 200 mg) only which are not sufficient to see the immunomodulatory effects.

4.What molecular pathways are involved in the dose-dependent immunomodulatory effects of C. chinense methanol extract?

5.There are so many typographical mistakes in the manuscript and need to be revised by native speaker.

6.The references are not properly arranged according to the Journal format and also cross checked with citation in the text.

Reviewer #5: In this study,a number of organic extracts of Capsicum chinense Jacq. (methanol, ethyl acetate and petroleum ether) have been studied for their immunomodulatory and DNAdamage-protective functions. Detailed explanations are provided for the study. The submission needs to be revised before it can be accepted.

1.In the materials & methods section, you mentioned defatting the powder sample using n-hexane, petroleum ether, ethyl acetate and methanol. But there is no more detail about this procedure. What ratio did you use those solvents and what is the volume of the reaction vessel? If you used a method of previously published work, I suggest you add a reference to this section.

2.The sentence in section 2.4.1 “The animals were placed in polycarbonate transparent rat cages (430 mm long, 290 mm wide, and 201 mm high) maintaining at temperatures ranging from (24 ± 2) ºC and a dark-light cycle lasting 12 h.” should be rewritten.

3.For overall, which extract has the best immunomodulatory effect methanol extract, ethyl acetate extract or petroleum ether extract?

7. PLOS authors have the option to publish the peer review history of their article (what does this mean? ). If published, this will include your full peer review and any attached files.

**Do you want your identity to be public for this peer review?** For information about this choice, including consent withdrawal, please see our Privacy Policy .

**Reviewer #3: **Yes: ** Dr. Aamir Mushtaq**

**Reviewer #4: No**

**Reviewer #5: No**

---

## [Author Response · Author response to Decision Letter 4]

8 Jan 2025

Reviewer #3: Review Report:

The manuscript titled "Capsicum chinense Jacq. fruit plays an immunomodulatory role in a cytokine attenuation and DNA damage protection" suggests that Capsicum chinense Jacq. fruit has potential health benefits by modulating the immune system. It implies the fruit can reduce the overproduction of cytokines, which are signaling proteins involved in inflammation. Additionally, the fruit may protect against DNA damage, highlighting its role in cellular protection and overall immune health. Overall, the manuscript appears to present a promising study with implications for both basic research and clinical practice which highlights its potential as a natural therapeutic agent for enhancing immune health and cellular resilience.

Note: I reviewed the file named (PONE-D-24-29500-R3) downloaded from journal online portal and all my comments are according to the line numberings of the pdf file “PONE-D-24-29500-R3”

Detailed Comments/Recommendations:

Title:

The title of the study, "Capsicum chinense Jacq. fruit plays an immunomodulatory role in a cytokine attenuation and DNA damage protection" appears to be apt and engaging. It effectively highlights the fruit's potential therapeutic significance, appealing to researchers and healthcare professionals interested in natural interventions for inflammation and cellular protection.

Abstract:

It will be more attractive for the readers if the authors may add up aims and objective of the study at the start of abstract in 2 lines.

Author’s Response - We sincerely appreciate your diligence in reviewing our manuscript. The authors have outlined the aims and objectives at the outset of the document abstract.

Page 4, line 3: Mention the animal (rats).

Author’s Response – We appreciate your feedback. The authors have referenced the rats’ species (Long Evans rats) utilized in the study.

Page 4, Line 6: Mention the dose at which immune response was suppressed.

Author’s Response- Thank you so much for your thoughtful review of our manuscript! We're happy to share that we've included information about the dose at which the immune response was suppressed.

It will be more appropriate if the authors may add a few numerical results in the abstract. It will be more impressive for readers.

Author’s Response: We sincerely appreciate your insightful comments. The authors have incorporated numerical results into the abstract, which will enhance its impact for the readers.

Key Words: Well written.

Author’s Response -Thank you so much.

Introduction:

Page 5, Lines 8-13: Sentence is too long; it will be more impressive if authors may split into portions for the better understandings of the reader.

Author’s Response: We sincerely appreciate your valuable feedback. We have divided the sentence into segments to enhance the reader's understanding.

Page 5, Line 15-18: The statement is confusing for the reader. It can be rewritten for the ease of reader.

Author’s Response: We sincerely appreciate your feedback. The authors have revised the statement to eliminate any potential confusion for the readers.

Page 6, Lines 4 to onward: Please add biography of Capsicum chinense Jacq, along with its habitat and cultivations etc.

Author’s Response- Thank you so much for your suggestion. In response, we have included the biography of Capsicum chinense Jacq.

Materials and Methods:

Page 7, Line 8: How much Capsicum chinense Jacq. Was obtained? Please mention its quantity.

Author’s Response- We extend my sincere gratitude for your invaluable observation and for the manner in which the authors have addressed your suggestions.

Page 8, Line 6: Please mention temperature and pressure adjusted in rotary evaporator.

Author’s Response- Much appreciated. We mentioned the temperature and pressure adjustments that occurred in the rotary evaporator.

Page 8, Line 18: Kindly mention humidity conditions, provided to animals in animal house.

Author’s Response- Thank you for your thoughtful reply! The authors highlighted the humidity conditions in the animals' house.

Page 8, Line 3: Please mention quantity of antigen administered to animals along with route of administration.

Author’s Response-We truly appreciate your keen observation! We're excited to let you know that we have taken your feedback into account and made the necessary adjustments.

Page 10, Line 3: Please insert reference for the methodology of immunization.

Author’s Response- In response to your suggestion, we happily included the reference for the methodology of immunization.

Pages 10-11: References are missing for all the methods mentioned on page 10 and 11.

Author’s Response-We’ve added the references just like you suggested!

Page 11, Line 20: Please mention magnification/resolution power at which slides were observed under microscope.

Author’s Response- We would like to express our sincere gratitude. The authors have indicated the resolution capacity of the microscope.

Page 15, Line 4: please insert reference.

Author’s Response-The authors inserted the reference.



Results:

Page 16, Line 12-14: Please write the sentence in past tense.

Author’s Response- We express our gratitude for your insightful observation. The sentence has been composed in the past tense.

All results are well expressed

Discussion:

Page 21, Line 17-18: give full form of bw.

Author’s Response-Thank you. We have written the full form of bw.

Over all the discussion is well written and quite satisfactory.

Conclusion:

It is well written. The authors are requested to add on future prospects of the study in the last of conclusion.

Author’s Response- We appreciate your valuable comment. The authors have included the future prospects of the study in the conclusion section.

References:

Page 33, Line 7: The page numbers are missing.

Author’s Response- The page number is included.

The authors are advised to review all the references for the strict adherence of the uniformity of Journal’s references style.

Author’s Response- Thank you for your guidance. We have thoroughly examined all the references to ensure strict adherence to the Journal’s referencing style.

Tables:

Table 2: Please add unit (g) in the column of weight change.

Author’s Response- Thank you. The authors have added the unit in Table 2.

Concluding Remarks:

The paper necessitates the aforementioned corrections, inclusive of rectifying any errors highlighted, coupled with a meticulous proofreading to refine its English language usage. Upon addressing these amendments, the paper will meet the required standards for acceptance.

Reviewer #4: The manuscript entitles as “Capsicum chinense Jacq. fruit plays an immunomodulatory role in a cytokine attenuation” is written by Saha et al., is on the role of Capsicum chinense’s fruits in cytokine attenuation. The manuscript is interesting and can be a new addition to the literature but needs to address some comments before acceptance.

1.The graphic abstract needs to be revised by adding the necessary images only to make it more concise and informative.

Author’s Response: We would like to express our sincere gratitude for your diligent efforts in reviewing our manuscript. We acknowledge the importance of your insightful comments and suggestions, which significantly contribute to the enhancement of our work. The authors have revised the graphical abstract by incorporating the necessary images solely to render it more concise and informative.

2.Did the authors isolate/identified chemical compounds from the study plant’s fruits and screen them for these biological activities. The activity of the extracts can be due to the synergistic effect of the mixture of compounds and pure compounds can be more prominent and effective.

Author’s Response: Thank you so much for your time and patience to review our manuscript. The authors indeed identified chemical compounds by GC-MS from the fruits of the plant and evaluated them for certain biological activities mentioned in another article by the authors. The DOI for that article is https://doi.org/10.1016/j.jff.2024.106103. Since most of the compounds exhibit antioxidant, anti-inflammatory, and antimicrobial properties, we can confidently assert that the effects of these extracts are significant and compelling. Regulating immune responses becomes especially important during periods of heightened oxidative stress, which can weaken immune cell activity and lead to a state of systemic inflammation known as oxinflammation. Phytochemicals extracted from fruit, which are high in antioxidants, may help restore the balance between antioxidants and oxidants and modulate the immune response.

3.Why did the authors select methanol extract and use two doses of high concentrations (100 and 200 mg) only which are not sufficient to see the immunomodulatory effects.

Author’s Response: Thank you so much for your valuable comments. In the author’s preceding article (https://doi.org/10.1016/j.jff.2024.106103), it is unequivocally demonstrated that among the three extracts (petroleum ether, ethyl acetate, and methanol), the methanolic extracts exhibited the most pronounced antioxidative effect and significantly reduced hyperglycemia induced by oxidative stress in comparison to the other two extracts. We chose methanolic extracts for this reason.

We conducted a baseline selection based on an acute toxicity(LD50) test to determine the doses, that guided our choice of concentration levels. The test animals received a methanolic extract of C. chinense (500–2000 mg/kg). Results showed that the highest dose tested did not lead to mortality. Moving forward, we will explore other concentration experiments to broaden our efforts.

4.What molecular pathways are involved in the dose-dependent immunomodulatory effects of C. chinense methanol extract?

Author’s Response: We greatly appreciate your insightful feedback. The authors have examined the molecular pathways relevant to the dose-dependent immunomodulatory effects of C. chinense methanol extract in the discussion section of the article. In this section, the authors further clarified the mechanism.

Phagocytic Activity of Neutrophils:

The extract augments the phagocytic activity of neutrophils, constituting an integral component of the cell-mediated immune response. This is evidenced by an elevated neutrophil count and their enhanced capacity to adhere and migrate towards pathogens. Such observations signify activating pathways associated with neutrophil recruitment and adhesion, potentially involving cytokines and other chemokines.

T-Cell Proliferation:

The extract stimulates T-cells by boosting the delayed-type hypersensitivity (DTH) response. This process probably engages the T-cell receptor (TCR) signaling pathway, which is essential for T-cell activation and cytokine production release.

Cytokine Release and Inflammatory Response:

The extract enhances cytokine release, thereby promoting the inflammatory response. This process may engage the nuclear factor-kappa B (NF-κB) and mitogen-activated protein kinase (MAPK) pathways, which govern the expression of cytokines and other immune factor mediators.

Immunoglobulin Production:

The extract promotes the production of immunoglobulins, indicating a role in B-cell activation pathways and class-switch recombination, possibly influenced by cytokines such as IL-6.

Major Histocompatibility Complex (MHC) Expression:

Amino acids in the extract facilitate the production of MHC complexes, which are crucial for antigen presentation. This process includes pathways such as the antigen-processing and presentation pathways regulated by MHC class I and II molecules.

Trace Element Utilization:

The trace elements within the extract support T-cell proliferation and the function of lytic enzymes. They might affect pathways such as the calcineurin-NFAT (nuclear factor of activated T-cells) pathway, which plays a vital role in T-cell activation.

Amino Acids and Immunomodulation:

The amino acids in the extract might help produce immunoglobulins and various immune mediators. This relates to protein biosynthesis pathways and their contribution to strengthening the immune response.

In conclusion, the immunomodulatory effects of the methanol extract derived from C. chinense are likely mediated through mechanisms that encompass T-cell activation, cytokine release, phagocytosis, immunoglobulin production, and the expression of major histocompatibility complex (MHC). Additionally, the involvement of trace elements and amino acids in facilitating these processes is also noteworthy.

5.There are so many typographical mistakes in the manuscript and need to be revised by native speaker.

6.The references are not properly arranged according to the Journal format and also cross checked with citation in the text.

Author’s Response: Thank you so much for your valuable observation. W addressed your suggestions.

Reviewer #5: In this study,a number of organic extracts of Capsicum chinense Jacq. (methanol, ethyl acetate and petroleum ether) have been studied for their immunomodulatory and DNAdamage-protective functions. Detailed explanations are provided for the study. The submission needs to be revised before it can be accepted.

1.In the materials & methods section, you mentioned defatting the powder sample using n-hexane, petroleum ether, ethyl acetate and methanol. But there is no more detail about this procedure. What ratio did you use those solvents and what is the volume of the reaction vessel? If you used a method of previously published work, I suggest you add a reference to this section.

2.The sentence in section 2.4.1 “The animals were placed in polycarbonate transparent rat cages (430 mm long, 290 mm wide, and 201 mm high) maintaining at temperatures ranging from (24 ± 2) ºC and a dark-light cycle lasting 12 h.” should be rewritten.

Author’s Response: I sincerely appreciate your invaluable observations and acknowledge your suggestions. Rewritten.

3.For overall, which extract has the best immunomodulatory effect methanol extract, ethyl acetate extract or petroleum ether extract?

Author’s Response: I genuinely value your insightful observations and appreciate your suggestions. Certainly, methanolic extracts display the most effective immunomodulatory properties.

---

## [Editor Report · Decision Letter 4]

13 Jan 2025

PONE-D-24-29500R4Capsicum chinense Jacq. fruit plays an immunomodulatory role in a cytokine attenuation and DNA damage protectionPLOS ONE

Dear Dr. Rahman,

Thank you for submitting your manuscript to PLOS ONE. After careful consideration, we feel that it has merit but does not fully meet PLOS ONE’s publication criteria as it currently stands. Therefore, we invite you to submit a revised version of the manuscript that addresses the points raised during the review process.

We look forward to receiving your revised manuscript.

Kind regards,

Sairah Hafeez Kamran, PhD

Academic Editor

PLOS ONE

Journal Requirements:

Additional Editor Comments:

The authors were repeatedly asked in prior revisions to include the composition of milk in the manuscript, but this comment was not addressed.

Please include the composition in the modification.

---

## [Author Response · Author response to Decision Letter 5]

14 Jan 2025

Dear Editor

Thank you so much for your attention to our manuscript. We would like to thank the reviewers for their valuable queries, suggestions, and recommendations on our manuscript. We have gone through the comments and addressed all those in a point-to-point manner as follows:

Reviewer #3: Review Report:

The manuscript is titled "Capsicum chinense Jacq. fruit plays an immunomodulatory role in cytokine attenuation and DNA damage protection" suggests that Capsicum chinense Jacq. fruit has potential health benefits by modulating the immune system. It implies the fruit can reduce the overproduction of cytokines, which are signaling proteins involved in inflammation. Additionally, the fruit may protect against DNA damage, highlighting its role in cellular protection and overall immune health. Overall, the manuscript appears to present a promising study with implications for both basic research and clinical practice which highlights its potential as a natural therapeutic agent for enhancing immune health and cellular resilience.

Note: I reviewed the file named (PONE-D-24-29500-R3) downloaded from journal online portal and all my comments are according to the line numberings of the pdf file “PONE-D-24-29500-R3”

Detailed Comments/Recommendations:

Title:

The title of the study, "Capsicum chinense Jacq. fruit plays an immunomodulatory role in cytokine attenuation and DNA damage protection" appears to be apt and engaging. It effectively highlights the fruit's potential therapeutic significance, appealing to researchers and healthcare professionals interested in natural interventions for inflammation and cellular protection.

Abstract:

It will be more attractive for the readers if the authors may add up aims and objective of the study at the start of abstract in 2 lines.

Author’s Response - We sincerely appreciate your diligence in reviewing our manuscript. The authors have outlined the aims and objectives at the outset of the document abstract.

Page 4, line 3: Mention the animal (rats).

Author’s Response – We appreciate your feedback. The authors have referenced the rats’ species (Long Evans rats) utilized in the study.

Page 4, Line 6: Mention the dose at which immune response was suppressed.

Author’s Response- Thank you so much for your thoughtful review of our manuscript! We're happy to share that we've included information about the dose at which the immune response was suppressed.

It will be more appropriate if the authors may add a few numerical results in the abstract. It will be more impressive for readers.

Author’s Response: We sincerely appreciate your insightful comments. The authors have incorporated numerical results into the abstract, which will enhance its impact for the readers.

Key Words: Well written.

Author’s Response -Thank you so much.

Introduction:

Page 5, Lines 8-13: Sentence is too long; it will be more impressive if authors may split into portions for the better understandings of the reader.

Author’s Response: We sincerely appreciate your valuable feedback. We have divided the sentence into segments to enhance the reader's understanding.

Page 5, Line 15-18: The statement is confusing for the reader. It can be rewritten for the ease of reader.

Author’s Response: We sincerely appreciate your feedback. The authors have revised the statement to eliminate any potential confusion for the readers.

Page 6, Lines 4 to onward: Please add biography of Capsicum chinense Jacq, along with its habitat and cultivations etc.

Author’s Response- Thank you so much for your suggestion. In response, we have included the biography of Capsicum chinense Jacq.

Materials and Methods:

Page 7, Line 8: How much Capsicum chinense Jacq. Was obtained? Please mention its quantity.

Author’s Response- We extend my sincere gratitude for your invaluable observation and for the manner in which the authors have addressed your suggestions.

Page 8, Line 6: Please mention temperature and pressure adjusted in rotary evaporator.

Author’s Response- Much appreciated. We mentioned the temperature and pressure adjustments that occurred in the rotary evaporator.

Page 8, Line 18: Kindly mention humidity conditions, provided to animals in animal house.

Author’s Response- Thank you for your thoughtful reply! The authors highlighted the humidity conditions in the animals' house.

Page 8, Line 3: Please mention quantity of antigen administered to animals along with route of administration.

Author’s Response-We truly appreciate your keen observation! We're excited to let you know that we have taken your feedback into account and made the necessary adjustments.

Page 10, Line 3: Please insert reference for the methodology of immunization.

Author’s Response- In response to your suggestion, we happily included the reference for the methodology of immunization.

Pages 10-11: References are missing for all the methods mentioned on page 10 and 11.

Author’s Response-We’ve added the references just like you suggested!

Page 11, Line 20: Please mention magnification/resolution power at which slides were observed under microscope.

Author’s Response- We would like to express our sincere gratitude. The authors have indicated the resolution capacity of the microscope.

Page 15, Line 4: please insert reference.

Author’s Response-The authors inserted the reference.



Results:

Page 16, Line 12-14: Please write the sentence in past tense.

Author’s Response- We express our gratitude for your insightful observation. The sentence has been composed in the past tense.

All results are well expressed

Discussion:

Page 21, Line 17-18: give full form of bw.

Author’s Response-Thank you. We have written the full form of bw.

Over all the discussion is well written and quite satisfactory.

Conclusion:

It is well written. The authors are requested to add on future prospects of the study in the last of conclusion.

Author’s Response- We appreciate your valuable comment. The authors have included the future prospects of the study in the conclusion section.

References:

Page 33, Line 7: The page numbers are missing.

Author’s Response- The page number is included.

The authors are advised to review all the references for the strict adherence of the uniformity of Journal’s references style.

Author’s Response- Thank you for your guidance. We have thoroughly examined all the references to ensure strict adherence to the Journal’s referencing style.

Tables:

Table 2: Please add unit (g) in the column of weight change.

Author’s Response- Thank you. The authors have added the unit in Table 2.

Concluding Remarks:

The paper necessitates the aforementioned corrections, inclusive of rectifying any errors highlighted, coupled with a meticulous proofreading to refine its English language usage. Upon addressing these amendments, the paper will meet the required standards for acceptance.

Reviewer #4: The manuscript entitled “Capsicum chinense Jacq. fruit plays an immunomodulatory role in a cytokine attenuation” is written by Saha et al., is on the role of Capsicum chinense’s fruits in cytokine attenuation. The manuscript is interesting and can be a new addition to the literature but needs to address some comments before acceptance.

1.The graphic abstract needs to be revised by adding the necessary images only to make it more concise and informative.

Author’s Response: We would like to express our sincere gratitude for your diligent efforts in reviewing our manuscript. We acknowledge the importance of your insightful comments and suggestions, which significantly contribute to the enhancement of our work. The authors have revised the graphical abstract by incorporating the necessary images solely to render it more concise and informative.

2.Did the authors isolate/identified chemical compounds from the study plant’s fruits and screen them for these biological activities. The activity of the extracts can be due to the synergistic effect of the mixture of compounds and pure compounds can be more prominent and effective.

Author’s Response: Thank you so much for your time and patience to review our manuscript. The authors indeed identified chemical compounds by GC-MS from the fruits of the plant and evaluated them for certain biological activities mentioned in another article by the authors. The DOI for that article is https://doi.org/10.1016/j.jff.2024.106103. Since most of the compounds exhibit antioxidant, anti-inflammatory, and antimicrobial properties, we can confidently assert that the effects of these extracts are significant and compelling. Regulating immune responses becomes especially important during periods of heightened oxidative stress, which can weaken immune cell activity and lead to a state of systemic inflammation known as oxinflammation. Phytochemicals extracted from fruit, which are high in antioxidants, may help restore the balance between antioxidants and oxidants and modulate the immune response.

3.Why did the authors select methanol extract and use two doses of high concentrations (100 and 200 mg) only which are not sufficient to see the immunomodulatory effects.

Author’s Response: Thank you so much for your valuable comments. In the author’s preceding article (https://doi.org/10.1016/j.jff.2024.106103), it is unequivocally demonstrated that among the three extracts (petroleum ether, ethyl acetate, and methanol), the methanolic extracts exhibited the most pronounced antioxidative effect and significantly reduced hyperglycemia induced by oxidative stress in comparison to the other two extracts. We chose methanolic extracts for this reason.

We conducted a baseline selection based on an acute toxicity(LD50) test to determine the doses, that guided our choice of concentration levels. The test animals received a methanolic extract of C. chinense (500–2000 mg/kg). Results showed that the highest dose tested did not lead to mortality. Moving forward, we will explore other concentration experiments to broaden our efforts.

4.What molecular pathways are involved in the dose-dependent immunomodulatory effects of C. chinense methanol extract?

Author’s Response: We greatly appreciate your insightful feedback. The authors have examined the molecular pathways relevant to the dose-dependent immunomodulatory effects of C. chinense methanol extract in the discussion section of the article. In this section, the authors further clarified the mechanism.

Phagocytic Activity of Neutrophils:

The extract augments the phagocytic activity of neutrophils, constituting an integral component of the cell-mediated immune response. This is evidenced by an elevated neutrophil count and their enhanced capacity to adhere and migrate towards pathogens. Such observations signify activating pathways associated with neutrophil recruitment and adhesion, potentially involving cytokines and other chemokines.

T-Cell Proliferation:

The extract stimulates T-cells by boosting the delayed-type hypersensitivity (DTH) response. This process probably engages the T-cell receptor (TCR) signaling pathway, which is essential for T-cell activation and cytokine production release.

Cytokine Release and Inflammatory Response:

The extract enhances cytokine release, thereby promoting the inflammatory response. This process may engage the nuclear factor-kappa B (NF-κB) and mitogen-activated protein kinase (MAPK) pathways, which govern the expression of cytokines and other immune factor mediators.

Immunoglobulin Production:

The extract promotes the production of immunoglobulins, indicating a role in B-cell activation pathways and class-switch recombination, possibly influenced by cytokines such as IL-6.

Major Histocompatibility Complex (MHC) Expression:

Amino acids in the extract facilitate the production of MHC complexes, which are crucial for antigen presentation. This process includes pathways such as the antigen-processing and presentation pathways regulated by MHC class I and II molecules.

Trace Element Utilization:

The trace elements within the extract support T-cell proliferation and the function of lytic enzymes. They might affect pathways such as the calcineurin-NFAT (nuclear factor of activated T-cells) pathway, which plays a vital role in T-cell activation.

Amino Acids and Immunomodulation:

The amino acids in the extract might help produce immunoglobulins and various immune mediators. This relates to protein biosynthesis pathways and their contribution to strengthening the immune response.

In conclusion, the immunomodulatory effects of the methanol extract derived from C. chinense are likely mediated through mechanisms that encompass T-cell activation, cytokine release, phagocytosis, immunoglobulin production, and the expression of major histocompatibility complex (MHC). Additionally, the involvement of trace elements and amino acids in facilitating these processes is also noteworthy.

5.There are so many typographical mistakes in the manuscript and need to be revised by native speaker.

6.The references are not properly arranged according to the Journal format and also cross checked with citation in the text.

Author’s Response: Thank you so much for your valuable observation. W addressed your suggestions.

Reviewer #5: In this study,a number of organic extracts of Capsicum chinense Jacq. (methanol, ethyl acetate and petroleum ether) have been studied for their immunomodulatory and DNAdamage-protective functions. Detailed explanations are provided for the study. The submission needs to be revised before it can be accepted.

1.In the materials & methods section, you mentioned defatting the powder sample using n-hexane, petroleum ether, ethyl acetate and methanol. But there is no more detail about this procedure. What ratio did you use those solvents and what is the volume of the reaction vessel? If you used a method of previously published work, I suggest you add a reference to this section.

2.The sentence in section 2.4.1 “The animals were placed in polycarbonate transparent rat cages (430 mm long, 290 mm wide, and 201 mm high) maintaining at temperatures ranging from (24 ± 2) ºC and a dark-light cycle lasting 12 h.” should be rewritten.

Author’s Response: I sincerely appreciate your invaluable observations and acknowledge your suggestions. Rewritten.

3.For overall, which extract has the best immunomodulatory effect methanol extract, ethyl acetate extract or petroleum ether extract?

Author’s Response: I genuinely value your insightful observations and appreciate your suggestions. Certainly, methanolic extracts display the most effective immunomodulatory properties.

Md Atiar Rahman PhD

Corresponding author

and

Professor

Department of Biochemistry and Molecular Biology

University of Chittagong

Bangladesh

Fellow, Bangladesh Academy of Sciences

ORCID: https://orcid.org/0000-0002-4902-8923

Researcher ID: https://publons.com/researcher/362002/rahman-md-atiar/

Scopus ID: https://www.scopus.com/authid/detail.uri?authorId=55457954600

LiveDNA: https://livedna.org/880.19039

Institutional Webpage: www.cu.ac.bd

Personal Webpage: www.am-npr.cu.ac.bd

Google scholar ID: https://scholar.google.com/citations?hl=en&user=Hrbt-8QAAAAJ

---

## [Editor Report · Decision Letter 5]

16 Jan 2025

PONE-D-24-29500R5Capsicum chinense Jacq. fruit plays an immunomodulatory role in a cytokine attenuation and DNA damage protectionPLOS ONE

Dear Dr. Rahman,

Thank you for submitting your manuscript to PLOS ONE. After careful consideration, we feel that it has merit but does not fully meet PLOS ONE’s publication criteria as it currently stands. Therefore, we invite you to submit a revised version of the manuscript that addresses the points raised during the review process.

**ACADEMIC EDITOR:****Dear Authors**In previous revisions, I repeatedly asked the authors to add the composition of milk in the text. Once more, please include the composition or give a scientific justification for its exclusion; otherwise, the manuscript can be rejected.

We look forward to receiving your revised manuscript.

Kind regards,

Sairah Hafeez Kamran, PhD

Academic Editor

PLOS ONE

Journal Requirements:

Additional Editor Comments:**Dear Authors**In previous revisions, I repeatedly asked the authors to add the composition of milk in the text. Once more, please include the composition or give a scientific justification for its exclusion; otherwise, the manuscript can be rejected.

---

## [Author Response · Author response to Decision Letter 6]

28 Jan 2025

Response to reviewers

Date: 19 January 2025

To

The Editor

Plos One

Dear Editor

Thank you so much for your valuable time and patience to handle our manuscript entitled “Capsicum chinense Jacq. fruit plays an immunomodulatory role in a cytokine attenuation and DNA damage protection” to your esteemed journal “Plos One”. We are indeed very grateful that you put your utmost efforts to improve our manuscript through repetitive review process. We have gone through the comments of the reviewers in this phase and addressed all those through a rigorous revision of the manuscript. Authors’ point-to-point responses are duly included in the submission system.

All the comments have been addressed in the previous revision, however; in this version we have received only the following comment which has been as addressed in the following way:

Additional Editor Comments:

Dear Authors

In previous revisions, I repeatedly asked the authors to add the composition of milk in the text. Once more, please include the composition or give a scientific justification for its exclusion; otherwise, the manuscript can be rejected.

Author’s response: Thank you so much for your suggestion. The authors have inserted the composition and source of milk in the discussion section. A clear justification of using the milk is also incorporated in the text. A table is as well added in the table section to detail the milk composition. Hope it will completely quench the queries of the reviewers/editors.

We do hope that the revised manuscript will be led to fall into the next phase of publication procedures.

Your further responses will be highly appreciated.

Md Atiar Rahman PhD

Corresponding author

and

Professor

Department of Biochemistry and Molecular Biology

University of Chittagong

Bangladesh

Fellow, Bangladesh Academy of Sciences

ORCID: https://orcid.org/0000-0002-4902-8923

Researcher ID: https://publons.com/researcher/362002/rahman-md-atiar/

Scopus ID: https://www.scopus.com/authid/detail.uri?authorId=55457954600

LiveDNA: https://livedna.org/880.19039

Institutional Webpage: www.cu.ac.bd

Personal Webpage: www.am-npr.cu.ac.bd

Google scholar ID: https://scholar.google.com/citations?hl=en&user=Hrbt-8QAAAAJ

---

## [Editor Report · Decision Letter 6]

30 Jan 2025

Capsicum chinense Jacq. fruit plays an immunomodulatory role in a cytokine attenuation and DNA damage protection

PONE-D-24-29500R6

Dear Dr. Rahman,

We’re pleased to inform you that your manuscript has been judged scientifically suitable for publication and will be formally accepted for publication once it meets all outstanding technical requirements.

Kind regards,

Sairah Hafeez Kamran, PhD

Academic Editor

PLOS ONE

---

## [Editor Report · Acceptance letter]

PONE-D-24-29500R6

PLOS ONE

Dear Dr. Rahman,

I'm pleased to inform you that your manuscript has been deemed suitable for publication in PLOS ONE. Congratulations! Your manuscript is now being handed over to our production team.

Kind regards,

on behalf of

Dr. Sairah Hafeez Kamran

Academic Editor

PLOS ONE
